# DIFFUSION GUIDED ADVERSARIAL STATE PERTURBATIONS IN REINFORCEMENT LEARNING

## ABSTRACT

Reinforcement learning (RL) systems, while achieving remarkable success across various domains, are vulnerable to adversarial attacks. This is especially a concern in vision-based environments where minor manipulations of high-dimensional image inputs can easily mislead the agent's behavior. To this end, various defenses have been proposed recently, with state-of-the-art approaches achieving robust performance even under large state perturbations. Upon closer investigation, however, we found that the effectiveness of the current defenses is due to a fundamental weakness of the existing $l_p$-norm constrained attacks, which can barely alter the semantics of the input even under a relatively large perturbation budget. In this work, we propose SHIFT, a novel diffusion-based state perturbation attack to go beyond this limitation. Specifically, we train a history-conditioned diffusion model, enhanced with policy guidance and realism detection to generate perturbed states that are semantically different from the true states while remaining realistic and history-aligned to avoid detection. Evaluations show that our attack effectively breaks existing defenses, including the most sophisticated ones, and significantly lowers the agent's cumulative reward in various Atari games by more than 50%. The results highlight the vulnerability of RL agents to semantics-aware adversarial perturbations, indicating the importance of developing more robust policies for safety-critical domains.

## 1 INTRODUCTION

Reinforcement learning (RL) has seen significant advancements in recent years, becoming a key area of machine learning. RL's ability to enable agents to learn optimal decision-making policies through interaction with dynamic environments has led to breakthroughs in various fields. Beginning from AlphaGo (Silver et al., 2016), RL-based systems show the ability to surpass human performance in complex games. Beyond gaming, RL is driving innovations in robotics, self-driving cars (Kiran et al., 2020), and industrial automation, where agents learn to navigate, manipulate, and interact autonomously.

However, RL is vulnerable to various types of attacks, such as reward and state perturbations, action space manipulations, and model inference and poisoning (Ilahi et al., 2020). Recent studies have shown that an RL agent can be manipulated by perturbing its observation (Huang et al., 2017; Zhang et al., 2020a) and reward signals (Huang & Zhu, 2019), and a well-trained RL agent can be confounded by a malicious opponent behaving unexpectedly (Gleave et al., 2020). In particular, a malicious agent can subtly manipulate the observations of a trained RL agent, resulting in a significant drop in performance and cumulative reward (Zhang et al., 2020a; Sun et al., 2021). Such attacks exploit vulnerabilities in the agent's perception systems, including sensors and communication channels, without needing to cause obvious disruptions. This susceptibility to minor perturbations raises major concerns, particularly for RL applications in security-sensitive and safety-critical environments.

There are several defenses trying to mitigate state perturbation attacks. SA-MDP (Zhang et al., 2020a) points out adding a regularization term in the loss function during the training stage can help train a smoother and more robust policy. WocaR-MDP (Liang et al., 2022) further improves this method by estimating a worst case reward under perturbation during training. ATLA (Zhang et al., 2021) trains the agent's policy and attacker's policy alternatively, utilizing the fact that when the agent's policy is fixed, finding the optimal attack policy is a Markov Decision Process (MDP) and can be solved by

RL. This method can derive a more robust agent policy but suffers from a prohibitive computational cost when the environment's state space is large, such as in Atari games with raw pixels as input. More recently, CAR-DQN (Li et al., 2024) shows that a variant of DQN using the Bellman Infinity error can further improve the policy's robustness. In addition to the pure training stage approaches mentioned above, diffusion models have been utilized to either recover the true state (YANG & Xu, 2024) or generate a belief about the true state (Sun & Zheng, 2024) from perturbed states to further improve robustness. In particular, DP-DQN (Sun & Zheng, 2024) derives a robust policy against large perturbations by integrating a pessimistically trained Q-function and diffusion-based belief modeling. State-of-the-art attacks such as PGD (Zhang et al., 2020a), MinBest (Huang et al., 2017), PA-AD (Sun et al., 2021), and the high-sensitivity direction attacks (Korkmaz, 2023) cannot compromise these more advanced defenses.

However, we found that current attacks share two major shortcomings when applied to environments with raw pixel images as input as in the case of Atari games. First, with the exception of Korkmaz (2023), current attacks usually restrict a perturbed state to be within an $\epsilon$-ball of the true state, measured using an $l_p$ norm, to avoid detection, which constrains the attacker's search space for generating semantics changing perturbations. Although the high-sensitivity direction attacks in Korkmaz (2023) are able to go beyond the $l_p$ norm constraint, they mainly target changes in visually significant but non-essential semantics (see our evaluation results in Appendix E). Second, they focus on improving attack performance while ignoring the temporal dependencies across states. For example, PGD uses gradient descent to generate noise, PA-AD uses RL to find the best perturbations, and Korkmaz (2023) utilizes high-sensitive directions, without considering the history. Thus, these attacks cannot easily modify the essential semantics of the image input while keeping it realistic and plausible. Consequently, the perturbed states generated by these attacks can be easily denoised with the help of a history-conditioned diffusion model, so they fail against those diffusion-based defenses.

With these two shortcomings in mind, we propose SHIFT(**S**tealthy **H**istory al**I**gned di**F**fusion a**T**tack), a novel semantics-aware attack method that goes beyond the traditional $l_p$ norm constraint. Our approach generates effective attacks that are consistent with the physical rules of the environment and the agent's previous observations to avoid detection by both humans and AI. In particular, we utilize a diffusion model with classifier-free guidance to approximate history-aligned state generation, which is further improved using classifier guidance to generate effective and realistic perturbed images. SHIFT can break all known defenses and significantly lower agents' cumulative reward in various Atari games. Our results highlight that RL agents with image input are vulnerable to semantics-aware adversarial perturbations, which has important implications when deploying them in sensitive domains. As a preliminary defense strategy, we show that a diffusion-based approach can significantly improve robustness when the agent can occasionally probe and observe the true states.

## 2 PRELIMINARY

### 2.1 REINFORCEMENT LEARNING (RL)

A reinforcement learning environment can be formulated as a Markov Decision Process (MDP), usually denoted as a tuple $\langle S, A, P, R, \gamma, \rho_0 \rangle$, where $S$ is the state space and $A$ is the action space. $P : S \times A \to \Delta(S)$ is the transition function of the MDP, where $P(s'|s, a)$ denotes the probability of moving to state $s'$ given the current state $s$ and action $a$. $R : S \times A \to \mathbb{R}$ is the reward function where $R(s, a) = \mathbb{E}(R_t | s_{t-1} = s, a_{t-1} = a)$ and $R_t$ is the reward in time step $t$. Finally, $\gamma$ is the discount factor and $\rho_0$ is the initial state distribution. An RL agent wants to maximize its cumulative reward $G = \Sigma_{t=0}^{T} \gamma^t R_t$ over a time horizon $T \in \mathbb{Z}^+ \cup \{\infty\}$, by finding a (stationary) policy $\pi : S \to \Delta(A)$, which can be either deterministic or stochastic. For any policy $\pi$, the state-value and action-value functions are two standard ways to measure how good $\pi$ is. For MDPs with a finite or countably infinite state space and a finite action space, there is a deterministic and stationary policy that is simultaneously optimal for all initial states $s$. For large and continuous state and action spaces, deep reinforcement learning (DRL) incorporates the powerful approximation capacity of deep learning into RL and has found notable applications in various domains.

### 2.2 STATE PERTURBATION ATTACKS IN RL

First introduced in Huang et al. (2017), a **state perturbation attack** is a test-stage attack targeting an RL agent with a well-trained policy $\pi$. We consider the worst-case scenario where the attacker has

access to a clean environment and the victim's policy $\pi$ and any other deployed defense mechanisms. Further, the attacker has access to the true states at real-time.

At each time step, the attacker observes the true state $s_t$ and generates a perturbed state $\tilde{s}_t$. The agent, however, only observes $\tilde{s}_t$ (and not $s_t$) and takes an action $a_t$ based on its policy $\pi(\cdot|\tilde{s}_t)$. It is important to note that the attacker only interferes with the agent's observed state and does not modify the underlying MDP. Consequently, the true state at the next time step is governed by the transition dynamics $P(s_{t+1}|s_t, \pi(\cdot|\tilde{s}_t))$.

Common attack objectives include minimizing the agent's long-term cumulative reward or enforcing it to take a sequence of target actions chosen by the attacker. In this work, we focus on the second objective where at each time step $t$, the attacker aims to generate a perturbed state $\tilde{s}_t$ such that the agent would have a higher chance to select an attacker-specified action $\bar{a}_t$, rather than the default action $\pi(\cdot|s_t)$, by altering the semantic meaning of the true state $s_t$. Note that the two objectives are closely related as the set of target actions can be chosen to minimize the victim's long-term return. However, as we show in the evaluation results, even if the target actions are chosen myopically, and only a moderate portion of them are successfully enforced, our attack can lead to a significant loss in long-term return, even in the presence of strong defenses.

Further, the attacker needs to remain stealthy to avoid immediate detection and achieve its long-term goal. To this end, previous state perturbation attacks (Zhang et al., 2020a; Sun et al., 2021) restrict the attacker's ability by a budget $\epsilon$, so that $\tilde{s}_t \in B_\epsilon(s_t)$ where $B_\epsilon(s_t)$ is the $l_p$ ball centered at $s_t$ for some norm $p$ (typically an $l_\infty$ norm is used). However, state-of-the-art diffusion-based defenses (Sun & Zheng, 2024) are able to mitigate the restricted attack even with a large $\epsilon$. Thus, in this work, we consider a novel attack similar to unrestricted adversarial examples in the supervised learning setting (Song et al., 2018) by removing this constraint. Instead, we measure the stealthiness of attacks using more intuitive and practical measures on the realism and history-alignment of the perturbed states, as discussed below. A detailed discussion on related work can be found in Appendix B.

## 2.3 DENOISING DIFFUSION PROBABILISTIC MODEL (DDPM)

Diffusion models, particularly Denoising Diffusion Probabilistic Models (DDPMs), have recently gained attention as generative models that iteratively reverse a predefined diffusion process to generate data from noise (Ho et al., 2020). A DDPM model consists of two phases: a forward diffusion process that gradually adds noise to the data and a reverse denoising process to recover the original data.

**Forward Process.** The forward process is a fixed Markov chain that progressively corrupts the data $\mathbf{x}_0$ over $T$ time steps by adding Gaussian noise. At each step, the data evolve according to $q(x_i \mid x_{i-1}) = \mathcal{N}(x_i; \sqrt{1-\beta_i}x_{i-1}, \beta_i\mathbf{I})$, where $\beta_i \in (0,1)$ controls the noise level at step $i$. After many steps, the data is transformed into near isotropic Gaussian noise and can be expressed as $q(x_{1:T} \mid x_0) = \prod_{i=1}^T q(x_i \mid x_{i-1}) = \prod_{i=1}^T \mathcal{N}(x_i; \sqrt{1-\beta_i}x_{i-1}, \beta_t\mathbf{I})$.

**Reverse Process.** The reverse process manages to recover the data $x_0$ from the noisy sample $x_T$. The reverse process is another Markov chain, parameterized by a neural network $\epsilon_\theta(x_i, i)$, which predicts the noise added to the data at each time step $i$ in the forward process, The reverse transition is modeled as:

$$p_\theta(x_{i-1} \mid x_i) = \mathcal{N}(x_{i-1}; \mu_\theta(x_i, t), \sigma_\theta^2(x_i, i)\mathbf{I}), \tag{1}$$

where $\mu_\theta$ is the predicted mean and $\sigma_\theta^2$ is the variance of the reverse distribution at each time step $i$.

**Training.** The training goal of DDPM is to learn a model $\epsilon_\theta(x_i, i)$ that predicts the noise added to a data point $x_0$ during the forward diffusion process. $\mu_\theta(x_i, i)$ in the reverse process is expressed in terms of the predicted noise $\epsilon_\theta(x_i, i)$:

$$\mu_\theta(x_i, i) = \frac{1}{\sqrt{1-\beta_i}}\left(x_i - \frac{\beta_i}{\sqrt{1-\bar{\alpha}_i}}\epsilon_\theta(x_i, i)\right), \tag{2}$$

where $\bar{\alpha}_i = \prod_{n=1}^i (1-\beta_n)$ is the cumulative product of $(1-\beta_i)$ over time steps.

The variance $\sigma_\theta^2(x_i, i)$ can be predicted through a neural network or set by a predetermined scheduler. In DDPM, $\sigma_\theta^2(x_i, i)$ is set according to a fixed schedule as $\sigma_i^2 = \beta_i$.

The training objective is to minimize the difference between the true noise $\epsilon$ and the noise predicted by the model $\epsilon_\theta$. This objective can be written as $\mathcal{L}_{\text{simple}} = \mathbb{E}_{x_0, i, \epsilon}\left[\|\epsilon - \epsilon_\theta(x_i, i)\|^2\right]$, where $\epsilon \sim \mathcal{N}(0, \mathbf{I})$

is the noise added during the forward process. By minimizing this loss, the model learns to iteratively remove noise from $x_i$, ultimately generating high-quality samples from the learned data distribution.

# 3 SHIFT - STEALTHY HISTORY ALIGNED DIFFUSION ATTACK

In this section, we introduce SHIFT, which is a novel state perturbation attack built upon diffusion models that combines the classifier-free and classifier guidance methods. Below, we first discuss the motivation for using diffusion models to generate perturbed states, where we also formulate the attack objectives by giving a novel characterization of a realistic, semantics-aware, and history-aligned attack, from a static view and a dynamic view, respectively (Section 3.1). We then discuss how to achieve these goals in Section 3.2.

## 3.1 MOTIVATIONS AND ATTACK OBJECTIVES

State-of-the-art perturbation attacks against image input (such as PGD, MinBest, and PA-AD) are performed by adding $l_p$-norm constrained noise to the input. Consequently, the perturbed states often fall outside the set of states that can be generated by the underlying MDP (determined by the game engine in Atari games). Consider the snapshots from the Atari Pong game shown in Figure 1, where Figure 1c generated by the PGD attack with an attack budget $\epsilon = \frac{15}{255}$ under $l_\infty$ norm can be easily distinguished from the true state in Figure 1b. On the other hand, smaller perturbations are ineffective in manipulating the agent's actions, especially in the presence of strong defenses. The main reason is that these attacks typically cannot alter the **semantic meaning** of the original state. As demonstrated in Figure 1c, even with a relatively large attack budget ($\epsilon = \frac{15}{255}$), the pong ball and the paddles in the game maintain their positions.

Our objective is to go beyond the current $l_p$ norm-constrained attacks to generate more powerful and stealthy state perturbations. To this end, a key observation is that a carefully designed diffusion model can enable more effective attacks by generating semantics-changing state perturbations (e.g., Figure 1e) to mislead the victim to choose a target action $\bar{a}_t$ that differs from the desired action, leading to significant performance loss. However, a naive application of diffusion models may lead to unrealistic output that violates the physical rules, as shown in Figure 1d with two balls in the Pong game and Figure 1i with three chickens across lanes in the Freeway game.

To generate realistic perturbed states that are semantically different from original states, we introduce the following definitions.

**Definition 1** (Valid States). The set of valid states $S^*$ of an MDP $\langle S, A, P, R, \gamma, \rho_0 \rangle$ is defined as: $S^* := \{s \in S \mid \exists \pi \in \Pi, d_\pi(s) > 0\}$, where $\Pi$ denotes the set of all possible (stationary) policies and $d_\pi(s)$ represents the stationary state distribution under policy $\pi$, from the initial distribution $\rho_0$.

In other words, $S^*$ consists of all states that can be reached by following an arbitrary policy $\pi$ from the initial distribution $\rho_0$. However, ensuring strict validity is intractable with limited amount of data (as in the case of diffusion models). Thus, we introduce the concept of **realistic states** as a more practical measure, based on the projection distance between a state $s$ and the set of valid states $S^*$.

**Definition 2** (Realistic States). A state $s$ is defined as realistic if its projection distance to the set of valid states $S^*$ is bounded by a threshold $\delta$. Formally, the set of realistic states $S^r$ is defined as: $S^r := \{s \in S \mid \|\text{Proj}_{S^*}(s) - s\|_2 \leq \delta\}$, where $\text{Proj}_{S^*}(s) = \arg\min_{s' \in S^*} \|s' - s\|_2$ is the projection of $s$ onto $S^*$, and $\delta$ is a predefined threshold.

We consider realistic states to be indistinguishable from valid states and pose realism as an objective of our approach. With the above definition, we further define a perturbed state $\tilde{s}$ to be semantically different from the original state if it satisfies the following condition.

**Definition 3** (Semantics-Changing States). A perturbed state $\tilde{s}$ is considered to be semantically different from the true state $s$ when $\text{Proj}_{S^*}(\tilde{s}) \neq s$.

The definition states that a perturbed state $\tilde{s}$ changes the semantic meaning of the true state $s$ when its projection point to the valid set $\text{Proj}_{S^*}(s)$ differs from $s$. For example, the perturbed states in Figures 1d and 1e are semantically different from the true state in Figure 1b as they both changed the location of the pong ball. As discussed in the next section and validated in the experiments, our

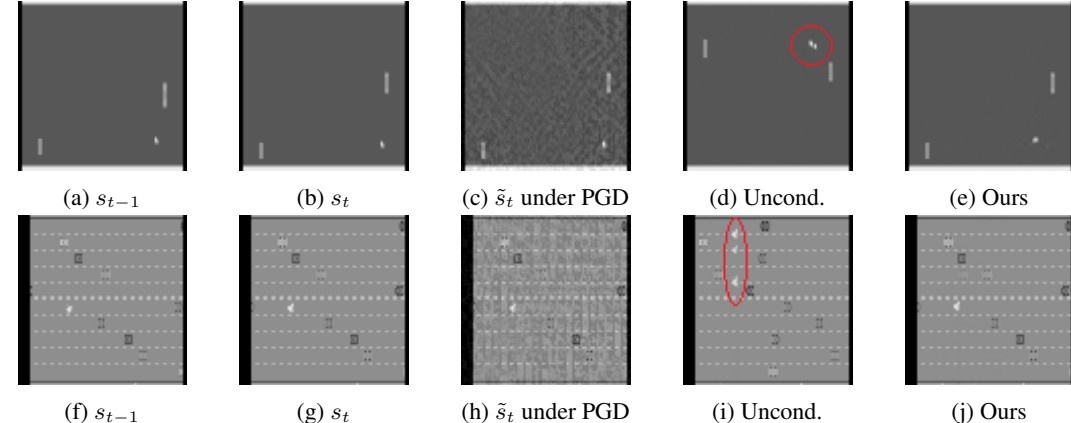

(a) $s_{t-1}$     (b) $s_t$     (c) $\tilde{s}_t$ under PGD     (d) Uncond.     (e) Ours

(f) $s_{t-1}$     (g) $s_t$     (h) $\tilde{s}_t$ under PGD     (i) Uncond.     (j) Ours

Figure 1: Examples of perturbed states in Atari Pong (first row) and Freeway (second row) games. The first two columns show the true states, the third column shows the perturbed states under the PGD attack with $l_\infty$ budget $\frac{15}{255}$, the fourth column shows the perturbed states generated by an unconditional diffusion model without realism and history guidance, and the last column shows the perturbed states generated by our method that are realistic and aligned with history.

attack is able to generate realistic perturbed states that are semantically different from the true states, which is the main reason why it can compromise state-of-the-art defenses.

While the above definitions capture the quality of individual states, they ignore the dynamic nature of sequential decision-making in RL and may lead to perturbed states that significantly deviate from history. For example, Figure 1d changes the semantic meaning of the original state, but it can be easily detected by inspecting the few states before it. Thus, we look for perturbed states that not only change the semantic meaning but also align with the history, formally defined as follows.

**Definition 4** (History-Aligned States). Let $H_{t-1} := (\mathrm{Proj}_{S^*}(\tilde{s}_{t-1}), a_{t-1}, \ldots, \mathrm{Proj}_{S^*}(\tilde{s}_{t-k}), a_{t-k})$ denote the sequence of last $k$ perturbed states observed by the victim (after projection onto $S^*$) and actions up to time $t$. Given the agent's policy $\pi$, a perturbed state $\tilde{s}_t$ at time step $t$ is aligned with $H_{t-1}$ from the agent's view if: $\tilde{s}_t \in S(H_{t-1}) := \{\tilde{s}_t \in S \mid \mathrm{Pr}_\pi(S_t = \mathrm{Proj}_{S^*}(\tilde{s}_t) \mid H_{t-1}) > 0\}$, where $S_t$ is the random variable for the true state at $t$.

That is, $\tilde{s}_t$ is aligned with the history if $\mathrm{Proj}_{S^*}(\tilde{s}_t)$ is a reachable next state given the victim's observed history up to time $t-1$, including the projection of past perturbed states onto $S^*$ along with their corresponding actions. This definition ensures that the perturbed state is undetectable even if the agent is equipped with a history-based detector. Note that instead of using the $\mathrm{Proj}_{S^*}(\cdot)$ operator, the above definition can be extended to incorporate the actual detector (if there is any) used by the agent.

We note that the above definition can be too restrictive in practice. In particular, for environments with deterministic transition functions (as in the case of Atari games), once we have $(s_{t-1}, a_{t-1})$, there is only one possible next state $s_t$. In this case, the historically aligned next state $\tilde{s}_t$ must satisfy $\mathrm{Proj}_{S^*}(\tilde{s}_t) = s_t$, leaving no space for attacks. To this end, we relax the definition as follows.

**Definition 5** (Approximately History-Aligned States). A perturbed state $\tilde{s}_t$ at time step $t$ is approximately aligned with a history $H_{t-1}$ if $\min_{s' \in S(H_{t-1})} \|\mathrm{Proj}_{S^*}(\tilde{s}_t) - s'\|_2 \leq \omega$. That is, we allow $\mathrm{Proj}_{S^*}(\tilde{s}_t)$ to deviate from $S(H_{t-1})$ by a threshold $\omega$.

As shown in Figure 1e, our approach can generate a perturbed state that slightly changes the pong ball's location to the left, which is approximately aligned with the given history.

## 3.2 DIFFUSION-BASED STATE PERTURBATIONS

In this section, we discuss SHIFT, our diffusion-based attack that can generate state perturbations to meet the three objectives defined above: semantics changing (towards the target action), realism, and approximate historical alignment. Our attack consists of two stages: the training stage and the testing stage. During the training stage, we use data generated by the clean environment (i.e., the MDP) to train a conditional diffusion model to generate states that are realistic and history-aligned. We further train an autoencoder to detect unrealistic states. In the testing stage, we employ the pretrained

diffusion model to generate perturbed states guided by (1) the defender's policy, which provides guidance toward the target action, and (2) the pre-trained autoencoder, which further enhances the realism of the perturbed states. Figure D.1 in the appendix illustrates the two stages of our attack and the main components involved, with each discussed in detail below.

While diffusion models have been utilized to generate adversarial examples in the supervised learning setting (see Appendix B.4 for a review), their application in adversarial state perturbations in RL has not been considered before. We remark that our problem can be viewed as sampling from a diffusion model with constraints on realism and history alignment. However, existing approaches for constrained diffusion (Christopher et al., 2024) cannot be directly applied to our setting as they require constraints such as physical rules to be explicitly given and easily evaluable. In our setting, it is difficult to identify the projection onto the valid states $S^*$, making these approaches less suitable.

### 3.2.1 GENERATING HISTORY-ALIGNED STATES VIA CONDITIONAL DIFFUSION

We start with a description of training the conditional diffusion model, which is built upon the classifier-free guidance approach (Ho & Salimans, 2022) that can generate both unconditional and conditional samples, enabling the model to guide itself during the generation process.

We train a classifier-free guidance model conditioned on a history to generate the next state $\tilde{s}_t$ that follows the given history. This ensures that the generated next state $\tilde{s}_t$ is realistic and aligned with the history, such that $\tilde{s}_t$ is stealthy from both the static and dynamic views. It is important to note that the true next state $s_t$ is independent of the victim's policy $\pi$ when the history (including previous states and actions) is given. Thus, we can train this diffusion model with classifier-free guidance without requiring knowledge of the specific victim's policy $\pi$. However, a separate diffusion model needs to be trained for each distinct MDP environment.

Specifically, let $\tau_{t-1} = \{s_{t-1}, a_{t-1}, ..., s_{t-k}, a_{t-k}\}$ be the **true** history from time $t - k$ to time $t$, where $k$ is a parameter. In our setting, the model is trained with both class-conditional data $(s_t, \tau_{t-1})$ and unconditional data $s_t$ by randomly dropping $\tau_{t-1}$ with a certain probability. The history $\tau_{t-1}$ and true state $s_t$ are sampled from trajectories generated in a clean environment by following a well-trained policy $\pi_{ref}$ (independent of the agent's policy) with exploration to ensure coverage. The noise prediction network $\epsilon_\theta(s_t^i, i, \tau_{t-1})$ is trained to learn both conditional and unconditional distributions during the training. When generating perturbed states at the testing stage, where the reverse process is applied, the noise prediction can be adjusted using a guidance scale $\Gamma(i)$ as follows:

$$\epsilon_i = \Gamma(i)\epsilon_\theta(s_t^i, i, \tau_{t-1}) + (1 - \Gamma(i))\epsilon_\theta(s_t^i, i), \tag{3}$$

where $\tau_{t-1}$ is the given history, and $\Gamma(i)$ controls the strength of the guidance. Note that we have two time step variables here, where $t$ is the time step in an RL episode and $i$ is the index of the reverse steps in the reverse process. Also, our attack uses the true history $\tau_{t-1}$ to approximate the victim's belief $H_{t-1}$, as the latter requires projecting each perturbed state onto $S^*$ and is computationally expensive. An interesting future direction is to incorporate the actual agent's belief modeling into diffusion-based generation to further improve our attack.

With classifier-free guidance, the model learns a distribution of $\tilde{s}_t$ conditioned on a historical trajectory $\tau_{t-1}$ during training with clean data pairs $(s_t, \tau_{t-1})$. Since the attacker has access to true states during the testing phase, they can set $\tau_{t-1}$ as a conditioning factor, forcing the generated perturbed state $\tilde{s}_t$ to align with the given historical trajectory $\tau_{t-1}$. Consequently, the classifier-free guidance enhances dynamic stealthiness, which aligns with our third attack objective on history alignment.

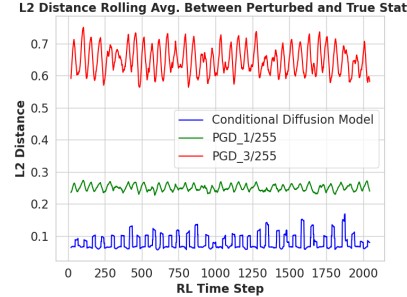

Figure 2: Distance between perturbed and true states ($84 \times 84$ grayscale images).

Since the classifier-free model is designed to generate the true next state $s_t$ based on the history $\tau_{t-1}$, the generated next state $\tilde{s}_t$ is expected to be close to the true state $s_t$ when the diffusion model is well-trained. Consequently, while the generated next state $\tilde{s}_t$ may not be exactly the same as $s_t$ to be classified as a valid state, $\tilde{s}_t$ is sufficiently close to $s_t$ to be considered as a realistic state according to Definition 2. This is confirmed in Figure 2, which shows the average $l_2$ distance between the perturbed states generated through the conditional diffusion model and the

true states in the Atari Freeway environment. Note that the $l_2$ distance gives an upper bound on the realism measure in Definition 2 as the true state may not be the closest state in $S^*$ with respect to the perturbed state. For comparison purposes, we also plot the distances for perturbed states generated by PGD with $l_\infty$ budget $\frac{1}{255}$ and $\frac{3}{255}$. It is shown that the states generated by the diffusion model conditioned on history are closer to the true states compared to states generated by the PGD method even with a small budget. This property enhances the realism of our generated perturbed states, satisfying our second attack objective.

### 3.2.2 GENERATING SEMANTICS CHANGING PERTURBATIONS VIA POLICY GUIDANCE

A perturbed state $\tilde{s}_t$ that is solely generated by the history-conditioned guidance discussed above is not able to manipulate the victim's action towards the target action, especially when an advanced defense method is deployed. To achieve our first attack objective of manipulating the victim's action, we introduce a classifier guidance module at the testing stage that can change the semantic meaning of the true state $s_t$. Classifier guidance is a method to improve the quality of samples generated by a diffusion model by incorporating class-conditional information (Dhariwal & Nichol, 2021). The core idea is to utilize a pre-trained classifier $p_\Phi(y|\mathbf{x})$, where $y$ represents the class label, to guide the reverse diffusion process toward generating samples conditioned on a desired class. In our context, we can treat the victim's policy $\pi$ as a classifier when the action space is discrete, which is the case for all Atari games considered in our evaluations.

Specifically, at each reverse time step $i$ of our pre-trained conditional diffusion model, the reverse process is modified by adjusting the mean of the noise prediction model $\epsilon_i$ with the gradient of the policy with respect to $\tilde{s}_t^i$ given the target action $\bar{a}_t$, that is, $\nabla_{\tilde{s}_t^i} \log \pi(\bar{a}_t | \tilde{s}_t^i)$. This guidance steers the generation process towards samples that are more likely to induce the victim to select the attacker-specified action $\bar{a}_t$, which deviates from the victim's default action $\pi(s_t)$, ultimately achieving the first attack objective and causing victim's performance loss at the same time. As shown in (Dhariwal & Nichol, 2021), for the unconditional reverse transition $p_\theta$ in (1), the modified reverse process with classifier guidance can be expressed as: $p(\tilde{s}_t^{i-1} \mid \tilde{s}_t^i, \bar{a}_t) = \mathcal{N}\left(\tilde{s}_t^{i-1}; \mu_\theta(\tilde{s}_t^i, i) + \sigma_i^2 \nabla_{\tilde{s}_t^i} \pi(\bar{a}_t \mid \tilde{s}_t^i), \sigma_i^2 \mathbf{I}\right)$.

In our scenario, however, classifier guidance is applied to a diffusion model conditioned on the history $\tau_{t-1}$. Typically, classifier guidance cannot be directly applied to a conditional diffusion model because the gradient term becomes $\nabla_{\tilde{s}_t^i} \pi(\bar{a}_t | \tilde{s}_t^i, \tau_{t-1})$, which is not easily computable through $\pi$. Fortunately, in our RL setting, the classifier guidance and classifier-free guidance can be combined as shown in the following theorem.

**Theorem 1.** The reverse process when sampling from a history-conditioned DDPM model guided by the victim's policy $\pi$ is given by $p(\tilde{s}_t^{i-1} \mid \tilde{s}_t^i, \bar{a}_t, \tau_{t-1}) = \mathcal{N}(\tilde{s}_t^{i-1}; \mu_i + \sigma_i^2 \nabla_{\tilde{s}_t^i} \log \pi(\bar{a}_t \mid \tilde{s}_t^i), \sigma_i^2 \mathbf{I})$, where $\mu_i$ is derived from $\epsilon_i$ in (3), as given by (2), and $\sigma_i^2$ is determined by the variance scheduler $\beta_i$.

Theorem 1 shows that classifier guidance and classifier-free methods can coexist without interference. While this is generally not true, it holds in our setting because given the two conditioning variables $\bar{a}_t$ and $\tau_{t-1}$, the noise predicted by classifier-free guidance depends only on $\tau_{t-1}$, while the gradient from classifier guidance depends solely on $\bar{a}_t$. A detailed proof is in Appendix C. Since the gradient information from classifier guidance modifies the reverse process, the generated perturbed state $\tilde{s}_t$, conditioned on $(\bar{a}_t, \tau_{t-1})$, will differ from states generated solely by conditioning on $\tau_{t-1}$. As a result, $\tilde{s}_t$ will be semantically distinct from the true state $s_t$, thus satisfying our first attack objective.

### 3.2.3 ENHANCING REALISM VIA AUTOENCODER GUIDANCE

Since the classifier guidance method introduces additional gradient information during the reverse process, the generated perturbed state $\tilde{s}_t$ may be less realistic. For example, in Figure 1d, two balls are simultaneously present in the Pong game, which alters the semantic meaning of the scene but is easily detectable by humans or automated anomaly detection tools. To mitigate this issue and improve the realism of the perturbed states, we incorporate an autoencoder-based anomaly detector trained on clean data. Autoencoders (Zhou & Paffenroth, 2017) are commonly used in unsupervised and semi-supervised anomaly detection tasks. They detect anomalies by measuring the reconstruction error, which is the difference between the input and its reconstruction. Since autoencoders are trained on normal data, they produce significantly higher reconstruction errors for anomalous inputs, as they have not learned to effectively encode or reconstruct these outliers.

In our approach, the attacker leverages training data sampled from a clean environment to train an autoencoder $\mathbf{AE}(\cdot)$ consisting of an encoder $\mathcal{E}_\phi(\cdot)$ and a decoder $\mathcal{D}_\psi(\cdot)$, with parameters $\phi$ and $\psi$, to detect unrealistic perturbed states during the testing stage. We define the reconstruction loss $\mathcal{L}$ as the $l_2$ distance between a state $s_t$ and the reconstructed state $\mathbf{AE}(s_t) = \mathcal{D}_\psi(\mathcal{E}_\phi(s_t))$, that is $\mathcal{L}(s_t, \mathbf{AE}(s_t)) = \|s_t - \mathbf{AE}(s_t)\|_2$, and train the autoencoder $\mathbf{AE}(\cdot)$ to minimize the average $l_2$ loss over the training set. Since input semantics vary significantly across environments, we need to train separate autoencoders for different environments. The pre-trained autoencoder is then used at the testing stage to improve the realism of the perturbed states, following the logic of classifier guidance. In particular, the attacker performs gradient descent at the end of each reverse step $i$, using the gradient of the reconstruction error with respect to $\tilde{s}_t^i$, to improve the realism of the generated state. This can be formulated as: $\tilde{s}_t^i = \tilde{s}_t^i - \nabla_{\tilde{s}_t^i} \mathcal{L}(\tilde{s}_t^i, \mathbf{AE}(\tilde{s}_t^i))$, where $\mathbf{AE}(\tilde{s}_t^i)$ represents the reconstructed sample, and $\mathcal{L}(\tilde{s}_t^i, \mathbf{AE}(\tilde{s}_t^i))$ is the reconstruction error. This process ensures that the generated perturbed state $\tilde{s}_t$ more closely aligns with realistic states, thus further supporting our second attack objective.

### 3.2.4 IMPLEMENTATION

Although our theoretical analysis in Section 3.2.2 is based on the DDPM method to simplify the discussion, we implement our attack method using the EDM formulation proposed in (Karras et al., 2022). EDM is a score-based diffusion method that efficiently guides the model through fewer reverse diffusion steps compared to DDPM, significantly reducing sampling time and allowing for real-time attacks in the testing stage. Previous work (Alonso et al., 2024) has successfully applied EDM based conditional diffusion model to world modeling of Atari environments, and we build our work upon their code. For the classifier-guidance implementation in the EDM model, we follow the technique in Ma et al. (2024) to weight joint and conditional guidance by two separate logits temperature parameters $\zeta_1$ and $\zeta_2$, which improves the generated sample quality. We further apply another technique used in (Bansal et al., 2024) to enhance the effectiveness of the classifier-guidance method. At each reverse step $i$, we first calculate the proposed output sample $\hat{s}_t$ based on the current state $\tilde{s}_t^i$ through the diffusion model without any attacks. Then we calculate the gradient of the victim's policy to guide the reverse process. These two techniques allow us to optimize the classifier guidance process more effectively, ensuring high-quality perturbed states that better satisfy the attack objectives. Details on the EDM formulation and implementation are provided in Appendix D, where both the training and the testing stage algorithms are given in Appendix D.5. In particular, the training of the history-conditioned EDM model is given in Algorithm 1. The implementation of the sampling process with both policy and realism guidance incorporated is given in Algorithm 2, where the autoencoder-based realism guidance appears in the last part of the algorithm.

## 4 EXPERIMENTS

In this section, we evaluate SHIFT using various Atari environments. We consider multiple state-of-the-art defenses including SA-DQN (Zhang et al., 2020a), WocaR-DQN (Liang et al., 2022), CAR-DQN (Li et al., 2024), and two diffusion-based defenses: Diffusion History YANG & Xu (2024), which is a test-stage defense, where the victim uses a diffusion model conditioned on perturbed history to recover true states, and DP-DQN (Sun & Zheng, 2024), which uses a diffusion-based denoiser on top of a pre-trained pessimistic policy. We set the history length $k = 4$ for the two history-based defenses and our attacks. All other hyper-parameters of  are given in Appendix D.6. Below, we discuss the main evaluation results, with additional results provided in Appendix E.

### 4.1 MAIN RESULTS

There are four key metrics in our experiments: **Reward**: the average episode return over 10 runs. **Manipulation Rate**: the percentage of RL time steps where the victim's real action $\tilde{a}_t = \pi(\tilde{s}_t)$ is the same as the target action $\bar{a}_t$. **Deviation Rate**: the percentage of RL time steps where the victim's real action $\tilde{a}_t$ differs from the default action $\pi(s_t)$. **Reconstruction Error**: the $l_2$ distance $\|\tilde{s}_t - \mathbf{AE}(\tilde{s}_t)\|_2$ between the perturbed state and the reconstructed state.

**Attack Performance.** We present the performance of our attack against various defense methods in four commonly used Atari environments in Table 1. To determine the target action $\bar{a}_t$, we utilize the Random Non-Optimal method, which randomly selects an action that differs from the victim's

| Env | Pong | | | Freeway | | |
|---|---|---|---|---|---|---|
| **Model** | **Reward** | **Manip. (%)** | **Dev. (%)** | **Reward** | **Manip. (%)** | **Dev. (%)** |
| **DQN-No Attack** | 21±0 | N/A | N/A | 34±0.1 | N/A | N/A |
| **DQN** | -20.7±0.5 | 87.1±1.9 | 89.6±1.7 | 0.1±0.3 | 41.8±1.5 | 54.8±1.4 |
| **SA-DQN** | -20.7±0.5 | 26.0±2.1 | 43.8±2.5 | 17.3±1.5 | 17.6±1.8 | 32.8±1.9 |
| **WocaR-DQN** | -20.4±0.8 | 22.2±1.4 | 40.9±1.9 | 22.1±0.0 | 12.9±0.8 | 25.3±1.5 |
| **CAR-DQN** | -20.6±0.5 | 47.4±2.2 | 72.4±2.9 | 18.4±0.8 | 27.3±1.3 | 35.2±1.5 |
| **DP-DQN** | 0.5±11.4 | 14.1±1.5 | 42.0±3.3 | 14.6±1.5 | 20.1±1.0 | 40.9±1.9 |
| **Diffusion History** | 6.0±6.2 | 8.4±0.5 | 25.3±0.9 | 19.1±1.2 | 13.8±1.0 | 26.9±1.2 |
| Env | BankHeist | | | RoadRunner | | |
| **Model** | **Reward** | **Manip. (%)** | **Dev. (%)** | **Reward** | **Manip. (%)** | **Dev. (%)** |
| **DQN-No Attack** | 680±0 | N/A | N/A | 13500±0 | N/A | N/A |
| **DQN** | 0±0 | 45.5±1.6 | 89.1±2.6 | 0±0 | 52.0±2.0 | 70.3±3.0 |
| **SA-DQN** | 14±9.2 | 20.1±0.7 | 50.4±2.6 | 260±215 | 34.1±2.3 | 54.2±1.2 |
| **WocaR-DQN** | 18±14.7 | 39.5±3.2 | 80.9±3.3 | 367±115 | 32.7±1.4 | 65.0±4.3 |
| **CAR-DQN** | 16±9.2 | 33.3±8.4 | 70.2±10.3 | 40±55 | 29.0±3.5 | 59.2±1.9 |
| **DP-DQN** | 2±4.2 | 15.0±3.2 | 77.0±13.3 | 360±321 | 12.1±1.0 | 55.8±2.5 |
| **Diffusion History** | 15±8.1 | 16.4±1.0 | 81.2±4.1 | 1480±788 | 9.1±2.0 | 43.1±2.1 |

Table 1: Episode reward, manipulation rate, and deviation rate of of SHIFT against various defense methods in different environments. All results are reported with mean and std over 10 runs.

| Pong | DDPM | | | EDM | | |
|---|---|---|---|---|---|---|
| | **Reward** | **Manip. (%)** | **Dev. (%)** | **Reward** | **Manip. (%)** | **Dev. (%)** |
| **DQN** | -20.6 ± 0.5 | 76.6 ± 1 | 83.6 ± 1 | -20.7 ± 0.5 | 87.1 ± 1.9 | 89.6 ± 1.7 |
| **Diffusion History** | 5.4 ± 5.6 | 15.1 ± 0.4 | 45.2 ± 0.3 | 6.0 ± 6.2 | 8.4 ± 0.5 | 25.3 ± 0.9 |
| **Sampling Time** | ~5 sec | | | ~0.2 sec | | |

Table 2: Efficiency and computational cost of DDPM vs. EDM diffusion architectures.

default action $\pi(s_t)$. The comparison of this method and a more advanced way of choosing target actions is provided in Appendix E.

Our results demonstrate that our attack significantly reduces the return of the vanilla DQN model and achieves a high rate of enforcing the target actions (i.e., a high manipulation rate) or non-optimal actions (i.e., a high deviation rate). When regularization-based defenses (SA-DQN, WocaR-DQN, and CAR-DQN) are employed, both the manipulation and deviation rates decline, yet they remain at levels that allow our attack to effectively compromise these defenses, resulting in low rewards across all environments. Further, SHIFT successfully circumvents diffusion-based defenses (Diffusion History and DP-DQN). Although these defenses perform better in the Pong environment due to their history-conditioned robust denoising capabilities, our attack can still bypass them because (1) the agent only has access to the perturbed history and (2) our attack is able to enforce semantic changes while being history aligned and maintaining low reconstruction errors and Wasserstein distance (see Figure 3a) , making it difficult to detect and mitigate.

**Comparison with Other Attacks.** We compare SHIFT with $l_\infty$-norm constrained PGD (Zhang et al., 2020a), temporally coupled PGD (Liang et al., 2024), MinBest (Huang et al., 2017) and PA-AD (Sun et al., 2021) attacks in the Atari Freeway environment in Figure 3a, We also include two high-sensitivity direction based attacks proposed in Korkmaz (2023). The PGD attack aims to force the victim into choosing non-optimal actions, while the MinBest attack seeks to minimize the logit of the best action. The temporally coupled PGD attack (PGD_TC) was adapted from Liang et al. (2024). The PA-AD attack uses RL to find best attack direction. We report the average return of PGD, MinBest, PA-AD attacks with varying attack budgets {3/255, 15/255}, PGD_TC with $\epsilon = 15/255$ ($\bar{\epsilon} = 7.5/255$), and two high-sensitivity direction based attacks, Blurred and Shifting (1,0), under the DP-DQN defense. We provide a comprehensive comparison and discussion with more high sensitivity direction based attacks in Appendix E. The results show that even with larger attack budgets, all these attacks fail to compromise the strong diffusion-based defense, as they can barely alter the essential semantic meaning of the states. In contrast, SHIFT succeeds in bypassing DP-DQN by generating semantics-changing perturbed states through policy guidance.

In Figure 3a, we also compare the average reconstruction loss of perturbed states and the average Wasserstein-1 distance between a perturbed state and the previous step's true state across a randomly sampled episode. The Wasserstein distance was proposed in Wong et al. (2019) as an alternative perturbation metric to $l_p$ distances, which measures the cost of moving pixel mass and can represent image manipulations more naturally than the $l_p$ distance. We argue that the reconstruction error captures static stealthiness of state perturbation while the Wasserstein distance to the previous state captures dynamic stealthiness. Our attack method achieves the best stealthiness from both

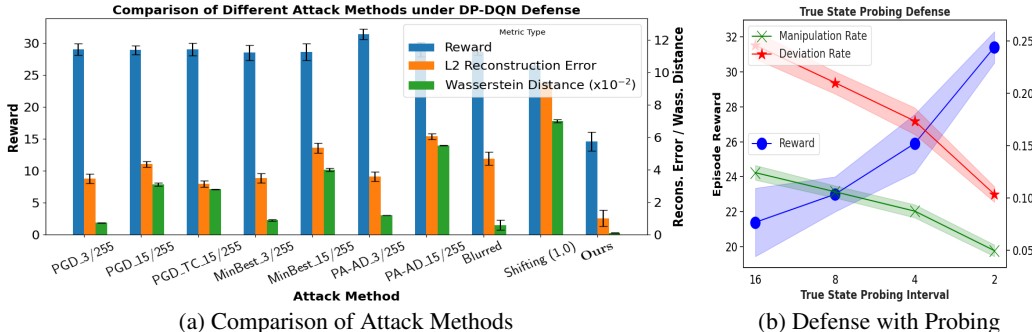

(a) Comparison of Attack Methods      (b) Defense with Probing

Figure 3: a) compares our attack with PGD, temporally correlated PGD, MinBest, PA-AD, Blurred, and Shifting attacks under different attack budgets against the DP-DQN defense. b) shows the performance of the Diffusion History defense under different probing intervals. All results are conducted in the Freeway environment.

perspectives according to Figure 3a. The superb attack performance and stealthiness brought by our method justifies the use of the conditional diffusion model to generate attacks.

In terms of time complexity, high-sensitivity direction attacks can generate perturbations instantaneously as they are policy independent, and PGD, PGD_TC, MinBest and PA-AD all take around 0.02 seconds to generate a perturbation with 10 iterations. Due to the computational overhead of the reverse process, diffusion-based methods typically require longer generation times. However, by adopting the EDM diffusion paradigm, we reduce the reverse process steps to 5, resulting in a generation time of approximately 0.2 seconds per perturbed state. Although slower than PGD, MinBest and PA-AD, this still allows our attack to remain feasible for real-time applications.

**Improving Robustness through Probing.** One potential solution to state-perturbation attacks is to allow the victim to probe the true states during the testing stage. For example, an autonomous car may query other cars or a central server to identify the true environment states, which, however, incurs a non-trivial cost and can only be done occasionally. To evaluate this idea, we enhance the Diffusion History defense by allowing the victim to probe true states at regular intervals, using a combination of historical true states and perturbed states to infer the current state. In Figure 3b, we present the performance of this probing strategy under different probing intervals in the Freeway environment. The results indicate that as the victim is allowed to probe the true states more frequently, the return improves, and both the manipulation and deviation rates decrease. A promising future direction is on guiding the victim to strategically probe, considering that some states are more critical than others.

**Ablation on DDPM and EDM diffusion architectures.** We compare DDPM and EDM in terms of attack efficiency and computational cost in Table 2. The results show that EDM and DDPM exhibit similar attack performance. However, DDPM is significantly slower than EDM in terms of sampling time (the average time needed to generate a single perturbed state during testing), making DDPM incapable of generating real-time attacks during testing. This validates the selection of EDM as the diffusion model architecture for constructing our attacks.

## 5 CONCLUSION AND LIMITATIONS

We introduce a novel diffusion-based state perturbation attack for reinforcement learning (RL) systems that extends beyond the traditional $l_p$-norm constraints. By leveraging conditional diffusion models, policy guidance, and realism enhancement techniques, we generate highly effective, semantically distinct, and stealthy attacks that cause a significant reduction in cumulative rewards across multiple Atari environments. Our results underscore the urgent need for more sophisticated defense mechanisms to effectively mitigate semantic uncertainties.

However, there are some limitations in our current attack method. First, the target action selection is myopic. A potential improvement could involve designing a joint planning-diffusion approach that determines target actions in a non-myopic manner. However, this requires evaluating the manipulation success rate of the diffusion model, which is computationally expensive. Second, although our method does not require training on separate defense policies, we still need to train individual conditional diffusion models for different environments. A promising direction is to enhance the transferability of the attack enabling a single conditional diffusion model to be effective across various environments.

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

APPENDIX

# A    BROADER IMPACTS

Our work introduces SHIFT, a novel approach to perturbation attacks in reinforcement learning (RL) by altering the semantics of the true states while remaining stealthy from both static and dynamic perspectives. SHIFT demonstrates outstanding performance, successfully compromising all state-of-the-art defense methods. This highlights the urgent need for more sophisticated defense mechanisms that are resilient to semantic uncertainties.

This research raises important safety concerns, as adversaries could exploit these semantic-changing attacks to cause significant harm in real-world RL applications, such as autonomous driving. The ability to alter the perception of critical systems like self-driving cars could lead to catastrophic consequences. As such, our findings underscore the necessity of further research into robust defenses capable of withstanding such advanced and subtle attack strategies.

# B    RELATED WORK

## B.1    STATE PERTURBATION ATTACKS AND DEFENSES

State perturbation attacks on RL policies were first introduced in Huang et al. (2017), where the *MinBest* attack was proposed to minimize the probability of selecting the best action by iteratively adding $l_p$-norm constrained noise calculated through $-\nabla_{\tilde{s}_t}\pi(\pi(\cdot|s_t), \tilde{s}_t)$. Building on this, Zhang et al. (2020a) showed that when the agent's policy is fixed, finding the optimal adversarial policy can be framed as an MDP, and the attacker can find the optimal attack policy by applying RL techniques. This was further improved in (Sun et al., 2021), where a more efficient algorithm for finding optimal attacks, called *PA-AD*, was introduced. Instead of searching perturbed states in the original state space, *PA-AD* trained a director through RL to find the optimal attack direction, and the trained director directs the designed actor to generate perturbed states, which decreases the searching space of RL. More recently, illusory attack (Franzmeyer et al., 2024) is proposed by requiring a perturbed trajectory to follow the same distribution as the normal trajectory, making it difficult to detect. However, this approach does not scale to high-dimensional image input. Korkmaz (2023) recognized the limitation of $l_p$ norm constrained attacks and proposed a policy-independent attack by following high sensitive directions, leading to attacks such as changing brightness and contrast, image blurring, image rotation and image shifting. These types of attacks are imperceptible when the amount of manipulation applied is small and can compromise SA-MDP defense. However, they can barely alter the essential semantics of image input according to our Definition 3. For example, in the Pong game, the pong ball's relative distance from the two pads will remain the same after changing brightness and contrast or shifting the image. Thus, as shown in Table 8 and Table 9, diffusion based defenses can protect the agent under these attacks.

On the defense side, Zhang et al. (2020a) demonstrated that a universally optimal policy under state perturbations might not always exist. They proposed a set of regularization-based algorithms (SA-DQN, SA-PPO, SA-DDPG) to train robust RL agents. This was enhanced in (Liang et al., 2022), where a worst-case Q-network and state importance weights were incorporated into the regularization. A more recent work called CAR-DQN (Li et al., 2024) shows using an $l_\infty$ norm can further improve the policy's robustness, and they theoretically capture the optimal robust policy (ORP) under $\epsilon$ constrained state perturbation attacks, although this method incurs high computational costs. Another line of work by Xiong et al. (2023) proposed an autoencoder-based detection and denoising framework to identify and correct perturbed states. Korkmaz & Brown-Cohen (2023) proposed SO-INRD, which uses the local curvature of the cross-entropy loss between the action distribution $\pi(a|s)$ and a targeted action distribution to detect adversarial directions. He et al. (2023b) showed that when the initial state distribution is known, it is possible to find a policy that optimizes the expected return under state perturbations. Diffusion-based defenses have also been utilized to generate more robust agent policies. DMBP (YANG & Xu, 2024) utilized a conditional diffusion model to recover actual states from perturbed states and Sun & Zheng (2024) used the diffusion model as a purification tool to generate a belief set about the actual state and perform a pessimistic training to generate a robust policy. More recently, a game-theoretical defense method (Grad) (Liang et al., 2024) was proposed to address temporally coupled attacks by modeling the temporally coupled robust RL problem as a partially

observable zero-sum game and deriving approximate equilibrium of the game. Another important recent defense is PROTECTED (Liu et al., 2024) that iteratively searches for a set of non-dominated policies during training and adapts these policies during testing to address different attacks. However, both Grad (Liang et al., 2024) and PROTECTED (Liu et al., 2024) focus on MuJoCo environments and are already computationally intensive (both take more than 20 hours) to train on the relatively simple environments. Without further adaptation, it will be computationally prohibitive to apply these two methods to Atari environments with image input.

## B.2 ATTACKS AND DEFENSES BEYOND STATE PERTURBATIONS

As demonstrated by (Huang & Zhu, 2019), altering the reward signal can significantly disrupt the training process of Q-learning, causing the agent to adopt a policy that aligns with the attacker's objectives. Additionally, (Zhang et al., 2020b) introduced an adaptive reward poisoning technique that can induce a harmful policy in a number of steps that scales polynomially with the size of the state space $|S|$ in the tabular setting. In a similar vein, Zhang et al. (2020b) developed an adaptive reward poisoning method capable of achieving a malicious policy in polynomial steps based on the size of the state space $|S|$.

Moving beyond reward manipulation, Lee et al. (2020) proposed two techniques for perturbing the action space. Among them, the *Look-Ahead Action Space* (LAS) method was found to deliver better performance in reducing cumulative rewards in deep reinforcement learning by distributing attacks across both the action and temporal dimensions. Another line of research focuses on adversarial policies within multi-agent environments. For example, (Gleave et al., 2020) showed that in a zero-sum game, a player using an adversarial policy can easily beat an opponent using a well-trained policy.

Attacks targeting an RL agent's policy network have also been explored. Inference attacks, as described by (Chen et al., 2021), aim to steal the policy network parameters. On the other hand, poisoning attacks, as discussed in (Huai et al., 2020), focus on directly manipulating the model parameters. Specifically, Huai et al. (2020) proposed an optimization-based method to identify an optimal strategy for poisoning the policy network.

## B.3 DIFFUSION MODELS AND RL

Diffusion models have recently been utilized to solve RL problems by exploiting their state-of-the-art sample generation ability. In particular, diffusion models have been utilized to generate high quality offline data in solving offline RL problems. Offline RL training is known as a data-sensitive process, where the quality of the data has a huge influence on the training result. To deal with this problem, many studies (He et al., 2023a; Ajay et al., 2022; Janner et al., 2022) have shown that diffusion models can learn from a demo dataset and then generate high reward trajectories for learning or planning purposes. In addition, conditional diffusion models have been directly used to model RL policies. A conditional diffusion model can generate actions through a denoising process with states and other useful information as conditions. Several studies (Kang et al., 2024; Wang et al., 2022) have shown state-of-the-art performance in various offline RL environments when using a diffusion model as a policy, which leads to a promising research direction.

Furthermore, Black et al. (2023) shows that the denoising process can be viewed as a Markov Decision Process (MDP). Thus, Black et al. (2023) trains a diffusion model with the help of RL by maximizing a user-specific reward function, which connects the generative models and optimization methods.

## B.4 DIFFUSION MODELS IN ADVERSARIAL EXAMPLES

Diffusion models have recently gained significant attention in generating adversarial examples due to their superb performances. They can generate high-quality adversarial examples that deceive target classifiers while remaining imperceptible to human observers.

Since the images generated by diffusion models inherently lack adversarial effects, a widely used approach is to use diffusion models along with existing methods of generating adversarial examples. The idea is to combine the generated samples from the diffusion model with perturbed samples from

other attack methods such as PGD attacks during the attack process to generate high quality and imperceptible adversarial examples (Xue et al., 2024; Chen et al., 2023b).

Another promising direction is to use a (surrogate) classifier to guide the diffusion model generating samples that meet attacker specified goals by using gradient information from the classifier during the testing stage (Liu et al., 2023; Dai et al., 2024; Guo et al., 2024). Also, Chen et al. (2023a) used the classifier guidance during the training stage of the diffusion model along with self and cross attention mechanisms.

Further, Beerens et al. (2024) showed that poisoning the training set can produce a deceptive diffusion model which will generate adversarial samples without any guidance.

However, these works only care about static stealthiness in a supervised learning setting, while SHIFT also takes dynamic stealthiness into consideration.

## C  PROOF OF THEOREM 1

The following proof is adapted from the proof in Appendix H of Dhariwal & Nichol (2021). We show that in the RL state perturbation attacks setting, we could combine classifier-free and classifier guidance. Let $\pi$ denote the victim's policy, $\bar{a}_t$ the target action at time $t$, and $\tau_{t-1} = \{s_{t-1}, a_{t-1}, ..., s_{t-k}, a_{t-k}\}$ the sequence of the last $k$ observations and actions up to time $t$. We first define a conditional Markovian process $\hat{q}$ similar to $q$ as follows.

$$
\begin{aligned}
\hat{q}(\tilde{s}_t^0 | \tau_{t-1}) &:= q(\tilde{s}_t^0 | \tau_{t-1}) \\
\hat{q}(\bar{a}_t | \tilde{s}_t^0, \tau_{t-1}) &\text{ is known for every } (\tilde{s}_t^0, \tau_{t-1}) \\
\hat{q}(\tilde{s}_t^{i+1} | \tilde{s}_t^i, \bar{a}_t, \tau_{t-1}) &:= q(\tilde{s}_t^{i+1} | \tilde{s}_t^i), \quad \forall i \\
\hat{q}\left(\tilde{s}_t^{1:T} \mid \tilde{s}_t^0, \bar{a}_t, \tau_{t-1}\right) &:= \prod_{i=1}^{T} \hat{q}\left(\tilde{s}_t^i \mid \tilde{s}_t^{i-1}, \bar{a}_t, \tau_{t-1}\right),
\end{aligned}
\tag{4}
$$

where $q(\tilde{s}_t^0 | \tau_{t-1}) = P(\tilde{s}_t^0 | s_{t-1}, a_{t-1})$ is the conditional distribution of the original state $\tilde{s}_t^0$ given the history $\tau_{t-1}$. Next we show that the joint distribution $\hat{q}(\tilde{s}_t^{0:T}, \bar{a}_t | \tau_{t-1})$ given $\tau_{t-1}$ is well defined.

$$
\begin{aligned}
\hat{q}(\tilde{s}_t^{0:T}, \bar{a}_t | \tau_{t-1}) &= \hat{q}\left(\tilde{s}_t^{1:T} \mid \tilde{s}_t^0, \bar{a}_t, \tau_{t-1}\right) \hat{q}(\tilde{s}_t^0, \bar{a}_t | \tau_{t-1}) \\
&= \prod_{i=1}^{T} \hat{q}\left(\tilde{s}_t^i \mid \tilde{s}_t^{i-1}, \bar{a}_t, \tau_{t-1}\right) \hat{q}(\bar{a}_t | \tilde{s}_t^0, \tau_{t-1}) \hat{q}(\tilde{s}_t^0 | \tau_{t-1}) \\
&= \prod_{i=1}^{T} \hat{q}\left(\tilde{s}_t^i \mid \tilde{s}_t^{i-1}, \bar{a}_t, \tau_{t-1}\right) \hat{q}(\bar{a}_t | \tilde{s}_t^0, \tau_{t-1}) \hat{q}(\tilde{s}_t^0 | \tau_{t-1}) \\
&= \prod_{i=1}^{T} \hat{q}\left(\tilde{s}_t^i \mid \tilde{s}_t^{i-1}, \bar{a}_t, \tau_{t-1}\right) \hat{q}(\bar{a}_t | \tilde{s}_t^0, \tau_{t-1}) \hat{q}(\tilde{s}_t^0 | \tau_{t-1}).
\end{aligned}
$$

Following essentially the same reasoning as in Appendix H of Dhariwal & Nichol (2021) with the trivial extension of including the condition $\tau_{t-1}$, we have

$$
\begin{aligned}
\hat{q}(\tilde{s}_t^i | \tilde{s}_t^{i-1}, \tau_{t-1}) &= \hat{q}(\tilde{s}_t^i | \tilde{s}_t^{i-1}) \\
\hat{q}(\tilde{s}_t^{i-1} | \tilde{s}_t^i, \tau_{t-1}) &= q(\tilde{s}_t^{i-1} | \tilde{s}_t^i, \tau_{t-1})
\end{aligned}
$$

Next, we show $\hat{q}(\bar{a}_t | \tilde{s}_t^i, \tilde{s}_t^{i-1}, \tau_{t-1})$ does not depend on $\tilde{s}_t^i$.

$$
\begin{aligned}
\hat{q}(\bar{a}_t | \tilde{s}_t^i, \tilde{s}_t^{i-1}, \tau_{t-1}) &= \frac{\hat{q}(\tilde{s}_t^{i-1}, \tilde{s}_t^i, \bar{a}_t, \tau_{t-1})}{\hat{q}(\tilde{s}_t^i, \tilde{s}_t^{i-1}, \tau_{t-1})} \\
&= \hat{q}(\tilde{s}_t^i | \tilde{s}_t^{i-1}, \bar{a}_t, \tau_{t-1}) \frac{\hat{q}(\tilde{s}_t^{i-1}, \bar{a}_t, \tau_{t-1})}{\hat{q}(\tilde{s}_t^{i-1}, \tilde{s}_t^i, \tau_{t-1})} \\
&= \hat{q}(\tilde{s}_t^i | \tilde{s}_t^{i-1}) \frac{\hat{q}(\bar{a}_t | \tilde{s}_t^{i-1}, \tau_{t-1})}{\hat{q}(\tilde{s}_t^i | \tilde{s}_t^{i-1}, \tau_{t-1})} \\
&= \hat{q}(\tilde{s}_t^i | \tilde{s}_t^{i-1}) \frac{\hat{q}(\bar{a}_t | \tilde{s}_t^{i-1}, \tau_{t-1})}{\hat{q}(\tilde{s}_t^i | \tilde{s}_t^{i-1})} \\
&= \hat{q}(\bar{a}_t | \tilde{s}_t^{i-1}, \tau_{t-1}).
\end{aligned}
\tag{5}
$$

We can now derive the reverse process that combines both classifier-free and classifier-guided methods.

$$
\begin{aligned}
\hat{q}(\tilde{s}_t^{i-1} | \tilde{s}_t^i, \bar{a}_t, \tau_{t-1}) &= \frac{\hat{q}(\tilde{s}_t^{i-1}, \tilde{s}_t^i, \bar{a}_t, \tau_{t-1})}{\hat{q}(\tilde{s}_t^i, \bar{a}_t, \tau_{t-1})} \\
&= \frac{\hat{q}(\tilde{s}_t^{i-1}, \tilde{s}_t^i, \tau_{t-1}) \hat{q}(\bar{a}_t | \tilde{s}_t^{i-1}, \tilde{s}_t^i, \tau_{t-1})}{\hat{q}(\tilde{s}_t^i, \bar{a}_t, \tau_{t-1})} \\
&= \frac{\hat{q}(\tilde{s}_t^{i-1} | \tilde{s}_t^i, \tau_{t-1}) \hat{q}(\tilde{s}_t^i, \tau_{t-1}) \hat{q}(\bar{a}_t | \tilde{s}_t^{i-1}, \tilde{s}_t^i, \tau_{t-1})}{\hat{q}(\tilde{s}_t^i, \bar{a}_t, \tau_{t-1})} \\
&= \frac{\hat{q}(\tilde{s}_t^{i-1} | \tilde{s}_t^i, \tau_{t-1}) \hat{q}(\bar{a}_t | \tilde{s}_t^{i-1}, \tilde{s}_t^i, \tau_{t-1})}{\hat{q}(\bar{a}_t | \tilde{s}_t^i, \tau_{t-1})} \\
&\overset{(a)}{=} q(\tilde{s}_t^{i-1} | \tilde{s}_t^i, \tau_{t-1}) \frac{\hat{q}(\bar{a}_t | \tilde{s}_t^{i-1}, \tau_{t-1})}{\hat{q}(\bar{a}_t | \tilde{s}_t^i, \tau_{t-1})},
\end{aligned}
$$

where (a) follows from Equation (5). Note that $q(\tilde{s}_t^{i-1} | \tilde{s}_t^i, \tau_{t-1})$ can be learned through a history conditioned diffusion model $p_\theta$ and we will use $\pi(\bar{a}_t | \tilde{s}_t^i)$ to approximate $\hat{q}(\bar{a}_t | \tilde{s}_t^i, \tau_{t-1})$, $\forall i$. We notice that the $\hat{q}(\bar{a}_t | \tilde{s}_t^i, \tau_{t-1})$ term guides our reverse process to generate samples that lead to a higher probability of choosing $\bar{a}_t$. Thus, the attacker should use the victim's policy $\pi$ here to gain better guidance toward $\bar{a}_t$. Plugging them back into the above equation, we have

$$
\begin{aligned}
\hat{q}(\tilde{s}_t^{i-1} | \tilde{s}_t^i, \bar{a}_t, \tau_{t-1}) &\approx p_\theta(\tilde{s}_t^{i-1} | \tilde{s}_t^i, \tau_{t-1}) \frac{\pi(\bar{a}_t | \tilde{s}_t^{i-1})}{\pi(\bar{a}_t | \tilde{s}_t^i)} \\
&\approx p_\theta(\tilde{s}_t^{i-1} | \tilde{s}_t^i, \tau_{t-1}) e^{\log \pi(\bar{a}_t | \tilde{s}_t^{i-1}) - \log \pi(\bar{a}_t | \tilde{s}_t^i)}.
\end{aligned}
\tag{6}
$$

Using the Taylor expansion, we get

$$
\log \pi(\bar{a}_t | \tilde{s}_t^{i-1}) - \log \pi(\bar{a}_t | \tilde{s}_t^i) \approx (\tilde{s}_t^{i-1} - \tilde{s}_t^i) \nabla_{\tilde{s}_t^i} \log \pi(\bar{a}_t | \tilde{s}_t^i).
$$

We also have

$$
p_\theta(\tilde{s}_t^{i-1} | \tilde{s}_t^i, \tau_{t-1}) \propto \mathcal{N}(\tilde{s}_t^{i-1}; \mu_i, i), \sigma_i^2 \mathbf{I}) \propto \exp\left( -\frac{\left(\tilde{s}_t^{i-1} - \mu_i\right)^2}{2\sigma_i^2} \right).
$$

where $\mu_i$ comes from $\epsilon_i = \Gamma \epsilon_\theta(s_t^i, i, \tau_{t-1}) + (1 - \Gamma) \epsilon_\theta(s_t^i, i)$, as given by (2), and $\sigma_i^2$ is determined by the noise scheduler $\beta_i$.

Substituting them back into (6), we have

$$p_\theta(\tilde{s}_t^{i-1}|\tilde{s}_t^i, \tau_{t-1})e^{\log\pi(\bar{a}_t|\tilde{s}_t^{i-1})-\log\pi(\bar{a}_t|\tilde{s}_t^i)}$$

$$\propto \exp\left(-\frac{\left(\tilde{s}_t^{i-1} - \mu_i\right)^2}{2\sigma_i^2} + \left(\tilde{s}_t^{i-1} - \tilde{s}_t^i\right)\nabla_{\tilde{s}_t^i}\log\pi(\bar{a}_t|\tilde{s}_t^i)\right)$$

$$= \exp\left(-\frac{\left(\tilde{s}_t^{i-1} - \mu_i\right)^2 - 2\sigma_i^2\left(\tilde{s}_t^{i-1} - \tilde{s}_t^i\right)\nabla_{\tilde{s}_t^i}\log\pi(\bar{a}_t|\tilde{s}_t^i)}{2\sigma_i^2}\right)$$

$$= \exp\left(-\frac{\left(\tilde{s}_t^{i-1} - \mu_i\right)^2 - 2\sigma_i^2\left(\tilde{s}_t^{i-1} - \mu_i\right)\nabla_{\tilde{s}_t^i}\log\pi(\bar{a}_t|\tilde{s}_t^i) + \left(\sigma_i^2\nabla_{\tilde{s}_t^i}\log\pi(\bar{a}_t|\tilde{s}_t^i)\right)^2}{2\sigma_i^2}\right)$$

$$\times \exp\left(\frac{2\sigma_i^2\left(\mu_i - \tilde{s}_t^i\right)\nabla_{\tilde{s}_t^i}\log\pi(\bar{a}_t|\tilde{s}_t^i) + \left(\sigma_i^2\nabla_{\tilde{s}_t^i}\log\pi(\bar{a}_t|\tilde{s}_t^i)\right)^2}{2\sigma_i^2}\right)$$

$$= \exp\left(-\frac{\left(\left(\tilde{s}_t^{i-1} - \mu_i\right) - \sigma_i^2\nabla_{\tilde{s}_t^i}\log\pi(\bar{a}_t|\tilde{s}_t^i)\right)^2}{2\sigma_i^2} + \frac{2\sigma_i^2\left(\mu_i - \tilde{s}_t^i\right)\nabla_{\tilde{s}_t^i}\log\pi(\bar{a}_t|\tilde{s}_t^i) + \left(\sigma_i^2\nabla_{\tilde{s}_t^i}\log\pi(\bar{a}_t|\tilde{s}_t^i)\right)^2}{2\sigma_i^2}\right)$$

$$\propto \exp\left(-\frac{\left(\tilde{s}_t^{i-1} - \left(\mu_i + \sigma_i^2\nabla_{\tilde{s}_t^i}\log\pi(\bar{a}_t|\tilde{s}_t^i)\right)\right)^2}{2\sigma_i^2}\right). \tag{7}$$

Equation (7) implies that the reverse process when sampling from a history-conditioned DDPM model guided by the victim's policy can be represented as

$$p(\tilde{s}_t^{i-1}|\tilde{s}_t^i, \bar{a}_t, \tau_{t-1}) = \mathcal{N}\left(\tilde{s}_t^{i-1}; \mu_i + \sigma_i^2\nabla_{\tilde{s}_t^i}\log\pi\left(\bar{a}_t \mid \tilde{s}_t^i\right), \sigma_i^2\mathbf{I}\right).$$

# D    Implementation Details and Algorithms

## D.1    Two Stage Attacks Pipelines

Figure 4 gives an overview of our two-stage diffusion-based attack including all the major components involved.

## D.2    Score-Based Diffusion Model

As shown in Song et al. (2020), a diffusion process $\{\boldsymbol{x}(i)\}_{i\in[0,T]}$ can be represented as the solution to a standard stochastic differential equation (SDE):

$$d\boldsymbol{x} = \boldsymbol{f}(\boldsymbol{x}, i)di + g(i)d\boldsymbol{w},$$

where $\boldsymbol{f}$ represents the drift coefficient, which models the deterministic part of the SDE and determines the rate at which the process changes over time on average. $g(i)$ is called the diffusion coefficient, which represents the random part of the SDE and determines the magnitude of the noise. Finally, $\boldsymbol{w}$ represents a Brownian motion so that $g(i)d\boldsymbol{w}$ is the noising process.

We can let the diffusion process have $\boldsymbol{x}_0 \sim p_0$ and $\boldsymbol{x}_T \sim p_T$, where $p_0 = p_{\text{data}}$ is the data distribution and $p_T$ is a Gaussian noise distribution independent of $p_0$. Then we could run the reverse-time SDE to recover a sample from $p_0$ by the following process:

$$d\boldsymbol{x} = \left[\boldsymbol{f}(\boldsymbol{x}, i) - g(i)^2\nabla_{\boldsymbol{x}}\log p_i(\boldsymbol{x})\right]di + g(i)d\overline{\boldsymbol{w}},$$

where $\nabla_{\boldsymbol{x}}\log p_i(\boldsymbol{x})$ is the score function and $\overline{\boldsymbol{w}}$ is a Brownian motion that flows back from time $T$ to 0. The training objective for the score matching fucntion $\boldsymbol{s}_\theta$ for the SDE is then given by:

$$\arg\min_\theta \mathbb{E}_i\left[\lambda(i)\mathbb{E}_{\boldsymbol{x}(0)}\mathbb{E}_{\boldsymbol{x}(i)|\boldsymbol{x}(0)}\left[\left\|\boldsymbol{s}_\theta(\boldsymbol{x}(i), i) - \nabla_{\boldsymbol{x}(i)}\log p_{0i}(\boldsymbol{x}(i) \mid \boldsymbol{x}(0))\right\|_2^2\right]\right],$$

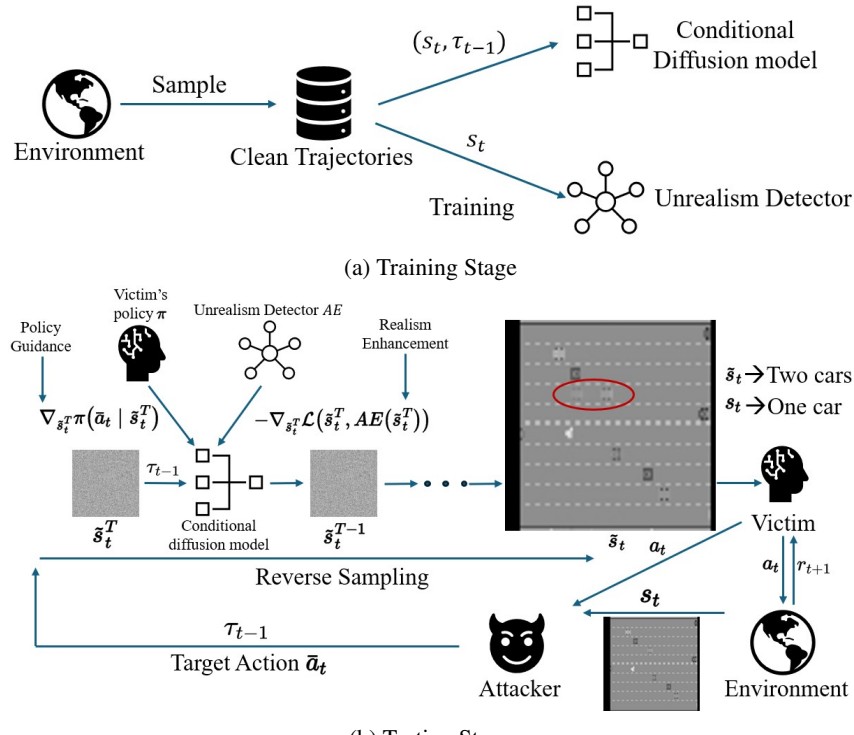

(a) Training Stage

(b) Testing Stage

Figure 4: Pipelines of SHIFT's two stages. a) shows the training stage where the attacker uses clean data to train a history-conditioned diffusion model and an autoencoder-based anomaly detector. b) shows the testing stage where the attacker perturbs the true state through the reverse sampling process of the pre-trained conditional diffusion model guided by the gradient of the victim's policy and that of the autoencoder's reconstruction loss.

where $\lambda(i)$ is a positive weighting function and $i$ is uniformly sampled from $[0, T]$. The objective can be further simplified since $p_{0i}$ is a known Gaussian distribution.

## D.3 EDM MODEL AND ODE FORMULATION

Inspired by Song et al. (2020), EDM (Karras et al., 2022) proposes an ODE formulation of the diffusion model by having a scheduler $\sigma(t)$ to schedule the noise added at each time step $t$. The score function correspondingly changes to $\nabla_{\boldsymbol{x}} \log p(\boldsymbol{x}; \sigma)$, which does not depend on the normalization constant of the underlying density function $p(\boldsymbol{x}, \sigma)$ and is much easier to evaluate. To be specific, if $D(\boldsymbol{x}; \sigma)$ is a denoiser function that minimizes:

$$\mathbb{E}_{\boldsymbol{x} \sim p_{\text{data}}} \mathbb{E}_{\boldsymbol{n} \sim \mathcal{N}(\mathbf{0}, \sigma^2 \mathbf{I})} \| D(\boldsymbol{x} + \boldsymbol{n}; \sigma) - \boldsymbol{x} \|_2^2 \tag{8}$$

then

$$\nabla_{\boldsymbol{x}} \log p(\boldsymbol{x}; \sigma) = (D(\boldsymbol{x}; \sigma) - \boldsymbol{x}) / \sigma^2$$

We usually train a neural network $\theta$ to learn the denoising function $D(\boldsymbol{x}, \sigma)$ by using the simplified training objective in Equation (8). Utilizing this finding, EDM only requires a small number of reverse sampling steps to generate a high quality sample. However, EDM needs more preconditioning parameters such as scaling $\boldsymbol{x}$ to an approximate dynamic range as further discussed below.

## D.4 EDM AS A CONDITIONAL DIFFUSION MODEL IN ATARI ENVIRONMENTS

In this paper, we follow the approach in Alonso et al. (2024) to train an EDM-based diffusion model conditioned on a history $\tau_{t-1}$ to predict the next state $s_t$. Taking the network preconditioning parameters used in EDM into account, we have the denoising function changed to:

$$\mathbf{D}_\theta \left( s_t^i, c_{\text{noise}}^i, \tau_{t-1} \right) = c_{\text{skip}}^i \, s_t^i + c_{\text{out}}^i \, \mathbf{F}_\theta \left( c_{\text{in}}^i \, s_t^i, c_{\text{noise}}^i, \tau_{t-1} \right),$$

where $\mathbf{F}_\theta$ is the neural network to be trained, the preconditioners $c_{\text{in}}$ and $c_{\text{out}}$ scale the network's input and output magnitude to keep them at unit variance for any noise level $\sigma(i)$, $c_{\text{noise}}^i$ is an empirical transformation of the noise level, and $c_{\text{skip}}^i$ is determined by the noise level $\sigma(i)$ and the standard deviation of the data distribution $\sigma_{\text{data}}$. The detailed expressions are given below:

$$c_{\text{in}}^i = \frac{1}{\sqrt{\sigma(i)^2 + \sigma_{\text{data}}^2}} \tag{9}$$

$$c_{\text{out}}^i = \frac{\sigma(i)\sigma_{\text{data}}}{\sqrt{\sigma(i)^2 + \sigma_{\text{data}}^2}} \tag{10}$$

$$c_{\text{noise}}^i = \frac{1}{4}\log(\sigma(i)) \tag{11}$$

$$c_{\text{skip}}^i = \frac{\sigma_{\text{data}}^2}{\sigma_{\text{data}}^2 + \sigma^2(i)} \tag{12}$$

where $\sigma_{\text{data}} = 0.5$. The noise parameter $\sigma(i)$ is sampled to maximize the effectiveness during training by setting $\log(\sigma(i)) = \mathcal{N}(P_{\text{mean}}, P_{\text{std}}^2)$, where $P_{\text{mean}} = -0.4, P_{\text{std}} = 1.2$. Refer to Karras et al. (2022) for a detailed explanation.

The training objective of $\mathbf{F}_\theta$ changes correspondingly to

$$\mathcal{L}(\theta) = \mathbb{E}[\|\mathbf{F}_\theta\left(c_{\text{in}}^i \, s_t^i, c_{\text{noise}}^i, \tau_{t-1}\right) - \frac{1}{c_{\text{out}}^i}\left(s_t - c_{\text{skip}}^i \, s_t^i\right)\|^2] \tag{13}$$

In our implementation, we change the residual block layers from [2,2,2,2] to [2,2] and the denosing steps to 5, and set drop conditions rate to 0.1. We keep other hyper-parameters the same as Alonso et al. (2024).

### D.5 TRAINING AND TESTING STAGE ALGORITHMS FOR DIFFUSION-BASED STATE PERTURBATIONS

---

**Algorithm 1:** History-Aligned Conditional Diffusion Model Training

---

**Input:** Training data $O = \{(s_t, \tau_{t-1})\}_{i=1}^N$, condition dropping rate $\alpha_{\text{drop}}$, $P_{\text{mean}}$, $P_{\text{std}}^2$, $\sigma$ , learning rate $\eta$
**Output:** Trained EDM model parameters $\theta$

**Initialize:** EDM model parameters $\theta$
**for** *number of training iterations* **do**
    Sample a data point $(s_t, \tau_{t-1}) \sim O$;
    Sample $\log(\sigma) \sim \mathcal{N}(P_{\text{mean}}, P_{\text{std}}^2)$;
    Calculate preconditioners $c_{\text{in}}, c_{\text{out}}$ based on $\sigma$ according to (9) and (10);
    Generate noisy data $s_t^i \sim \mathcal{N}(s_t, \sigma^2\mathbf{I})$;
    **if** *random* $> \alpha_{drop}$ **then**
        | Compute generated state $\tilde{s}_t = \mathbf{D}_\theta(s_t^i, c_{\text{noise}}, \tau_{t-1})$
    **else**
        | Compute generated state $\tilde{s}_t = \mathbf{D}_\theta(s_t^i, c_{\text{noise}})$
    Compute loss $\mathcal{L}(\theta)$ based on Equation (13);
    Update $\theta$ using gradient descent: $\theta \leftarrow \theta - \eta \cdot \nabla_\theta \mathcal{L}(\theta)$;
**end**

---

---

**Algorithm 2:** Testing Stage Sampling with History, Policy and Realism Guidance

---

**Input:** History conditioned diffusion model $\mathbf{D}_\theta$, victim's policy $\pi$, number of denoising steps $T$, autoencoder-based unrealism detector $\mathbf{AE}$, target attack action $\bar{a}_t$, given history $\tau_{t-1}$, joint logit temp $\zeta_1$, marginal logit temp $\zeta_2$, classifier-free guidance strength $\Gamma_1$, classifier guidance strength $\Gamma_2$

**Output:** Generated sample $\tilde{s}_t$

**Initialize:** $\tilde{s}_t^T \sim \mathcal{N}(0, \mathbf{I})$;
**for** $i = T$ **to** $1$ **do**
    // Calculate proposed output $\hat{s}_t$ based on $\tilde{s}_t^i$
    $\hat{s}_t = \tilde{s}_t^i$ ;
    **for** $j = i$ **to** $1$ **do**
        $\hat{s}_t = \mathbf{D}_\theta(\hat{s}_t, c_{\text{noise}}^j, \tau_{t-1})$
    **end**
    Policy guidance gradient $g \leftarrow \nabla_{\hat{s}_t} \log\left(\exp\left(\pi\left(\bar{a}_t|\hat{s}_t\right)\zeta_1\right) / \left(\sum_{a \in A} \exp\left(\pi\left(a|\hat{s}_t\right)\zeta_2\right)\right)\right)$;
    Inject policy guidance $\tilde{s}_t^i = \tilde{s}_t^i + \Gamma_2\left(g/\|g\|_2\right)$;
    Generate next sample $\tilde{s}_t^{i-1} = \Gamma_1(i)\mathbf{D}_\theta(\tilde{s}_t^i, c_{\text{noise}}^i, \tau_{t-1}) + (1 - \Gamma_1(i))\mathbf{D}_\theta(\tilde{s}_t^i, c_{\text{noise}}^i)$;
    **if** $i \neq 1$ **then**
        Conduct a gradient descent based on the reconstruction error from the unrealism detector
$$\tilde{s}_t^{i-1} = \tilde{s}_t^{i-1} - \nabla_{\tilde{s}_t^{i-1}}\mathcal{L}(\tilde{s}_t^{i-1}, \mathbf{AE}(\tilde{s}_t^{i-1}));$$
    **end**
**end**

---

### D.6 HYPER-PARAMETERS SETTING

**EDM Diffusion Model Training Parameters.** As mentioned before, we only change the residual block layers from [2,2,2,2] to [2,2] and the denosing steps to $5$, and set drop conditions rate to $0.1$. We keep other hyper-parameters the same as Alonso et al. (2024) for training the EDM diffusion model.

**Testing Stage Parameters and Testbench Specification.** We use the same logit temperatures as Ma et al. (2024) with $\zeta_1 = 1.0$ and $\zeta_2 = 0.0$. We schedule the classifier-free guidance scale as $\Gamma_1(i) = \max(\frac{T-i}{T}, 0.3)$, where $T$ is the number of reverse steps and $i$ is the current reverse step. We set the policy guidance strength $\Gamma_2$ differently in each environment under each defense. In the Pong environment, we set $\Gamma_2 = 3.5$ for DQN, DP-DQN and Diffusion History and $\Gamma_2 = 2$ for all other defenses. In the Freeway environment, we set $\Gamma_2 = 6$ for DQN, DP-DQN and Diffusion History and $\Gamma_2 = 4.5$ for all other defenses. In the BankHeist environment, we set $\Gamma_2 = 4$ for all defenses.

We conduct all of our experiments on a workstation equipped with an Intel I9-12900KF CPU, an RTX 3090 GPU, and 64GB system RAM.

**Atari Environments Pre-processing.** We have used the same environment wrappers as in Zhang et al. (2020a), which convert a RGB image to a gray-scale image and resize the image to reduce its resolution from $210 \times 160$ to $84 \times 84$. We also follow Zhang et al. (2020a) to center crop images using the same shifting parameters as in Zhang et al. (2020a), where we set the cropping shift to 10 for Pong, 20 for Roadrunner, and 0 for Freeway and Bankhesit. We do not stack frames in our pre-processing.

## E MORE EVALUATION RESULTS

**Realism Guidance.** Figure 5a illustrates the $l_2$ reconstruction error, defined as $\|s_t - \mathbf{AE}(s_t)\|_2$, for generated perturbed states both with and without the realism enhancement component. The figure demonstrates that, by incorporating realism enhancement, the $l_2$ reconstruction error is significantly reduced. This reduction indicates that realism enhancement effectively contributes to the generation of perturbed states that are more stealthy and less likely to be detected.

| Env | Pong (Random Non-Optimal) | | | Pong (Min Q) | | |
|---|---|---|---|---|---|---|
| Model | Reward | Manipulation Rate | Deviation Rate | Reward | Manipulation Rate | Deviation Rate |
| DQN-No Attack | 21±0 | N/A | N/A | 21±0 | N/A | N/A |
| DQN | -20.7±0.5 | 87.1% ± 1.9% | 89.6% ± 1.7% | -20.0±0.0 | 79.7.1% ± 2.8% | 84.3% ± 2.5% |
| SA-DQN | -20.7±0.5 | 26.0% ± 2.1% | 43.8% ± 2.5% | -20.9±0.3 | 19.8% ± 3.7% | 44.0% ± 5.0% |
| WocaR-DQN | -20.4±0.8 | 22.2% ± 1.4% | 40.9% ± 1.9% | -21±0 | 17.9% ± 0.4% | 33.6% ± 0.8% |
| CAR-DQN | -20.6±0.5 | 47.4% ± 2.2% | 72.4% ± 2.9% | -20.9±0.3 | 46.5% ± 3.4% | 73.8% ± 2.3% |
| DP-DQN | 0.5±11.4 | 14.1% ± 1.5% | 42.0% ± 3.3% | 0.5±11.9 | 6.2% ± 2.0% | 43.7% ± 3.4% |
| Diffusion History | 6.0±6.2 | 8.4% ± 0.5% | 25.3% ± 0.9% | -11.5±4.8 | 7.0% ± 0.4% | 30.5% ± 2.0% |

Table 3: Ablation results for different target action selection methods

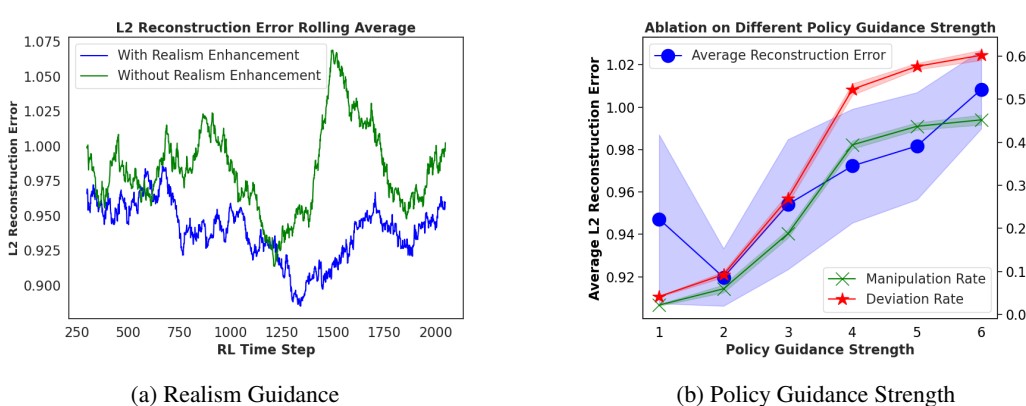

(a) Realism Guidance

(b) Policy Guidance Strength

Figure 5: Ablation Study Results. a) shows the rolling average of $l_2$ reconstruction error (from the autoencoder-based realism detector) of our generated perturbed states with and without the realism enhancement. b) shows the $l_2$ reconstruction error, manipulation rate and deviation rate under different policy guidance strengths. (a) and (b) use the vanilla DQN policy.

**Selection of Target Actions.** Table 3 presents a comparison between two methods for selecting the target action $\bar{a}_t$: the Random Non-Optimal method and the Min Q method. In the Min Q method, the target action is defined as the action that minimizes the $Q$ value under the current state: $\bar{a}_t = \arg\min_{a \in A} Q_\pi(s_t, a)$. From the table, we observe that the Min Q method achieves comparable or improved attack performance across various defense methods. However, the manipulation rate is significantly lower than that of the Random Non-Optimal method. This discrepancy can be attributed to the increased difficulty of manipulating the victim into selecting the worst action, as the attacker must exert more effort to perturb the states in such a way that amplifies the logit of the worst action. Therefore, there exists a trade-off between the different methods for choosing target actions. Further, an attacker may employ a joint planning-diffusion strategy to minimize the victim's expected return under a diffusion attack. However, this approach requires more detailed information about the manipulation success rate of a diffusion-based attack, which is nontrivial to obtain, and we consider it a potential future direction.

**Policy Guidance Strength.** Figure 5b illustrates the performance of our attack across various levels of policy guidance strength $\Gamma_2$. The figure indicates that as the strength increases, the effectiveness of our attack improves, leading to higher manipulation and deviation rates. However, this increased strength also results in a higher $l_2$ reconstruction error, which negatively impacts the realism of the generated perturbed states. Consequently, there exists a trade-off between attack effectiveness and stealthiness when selecting different policy guidance strengths.

| RoadRunner | Reward | Manipulation | Deviation |
|---|---|---|---|
| No Attack | 13500 ± 0 | NA | NA |
| DQN | 0 ± 0 | 52% ± 2% | 70% ± 3% |
| SA-DQN | 260 ± 215.41 | 34% ± 2% | 54% ± 1% |
| Diffusion History | 1480 ± 788.42 | 9% ± 2% | 43% ± 2% |

Table 4: SHIFT performance on the RoadRunner game under different defense methods.

| Freeway | PGD-1/255 | | PGD-3/255 | | PGD-15/255 | | MinBest-1/255 | | MinBest-3/255 | |
|---|---|---|---|---|---|---|---|---|---|---|
| | Reward | Dev (%) | Reward | Dev (%) | Reward | Dev (%) | Reward | Dev (%) | Reward | Dev (%) |
| DQN | $0 \pm 0$ | $86.2 \pm 0.5$ | $0 \pm 0$ | $100 \pm 0$ | $0 \pm 0$ | $100 \pm 0$ | $0 \pm 0$ | $100 \pm 0$ | $0 \pm 0$ | $100 \pm 0$ |
| SA-DQN | $30 \pm 0$ | $0 \pm 0$ | $30 \pm 0$ | $0 \pm 0$ | $20 \pm 1.6$ | $8 \pm 10$ | $30 \pm 0$ | $0 \pm 0$ | $29 \pm 1.4$ | $0.4 \pm 0.3$ |
| DP-DQN | $30 \pm 0.9$ | $3.5 \pm 0.2$ | $30 \pm 0.9$ | $4.5 \pm 0.3$ | $29 \pm 1$ | $3.2 \pm 0.1$ | $30.2 \pm 1.3$ | $3.7 \pm 0.3$ | $30.6 \pm 1.4$ | $4.1 \pm 0.1$ |
| | MinBest-15/255 | | PA-AD-1/255 | | PA-AD-3/255 | | PA-AD-15/255 | | Ours | |
| | Reward | Dev (%) | Reward | Dev (%) | Reward | Dev (%) | Reward | Dev (%) | Reward | Dev (%) |
| DQN | $0 \pm 0$ | $100 \pm 0$ | $0 \pm 0$ | $100 \pm 0$ | $0 \pm 0$ | $100 \pm 0$ | $0 \pm 0$ | $100 \pm 0$ | $\mathbf{0.1 \pm 0.3}$ | $\mathbf{54 \pm 1.4}$ |
| SA-DQN | $20.8 \pm 2.5$ | $9.1 \pm 1.1$ | $30 \pm 0$ | $0 \pm 0$ | $30 \pm 0$ | $0 \pm 0$ | $20.5 \pm 4.4$ | $3 \pm 1$ | $\mathbf{17.3 \pm 1.5}$ | $\mathbf{33 \pm 2}$ |
| DP-DQN | $29.4 \pm 1.2$ | $7.3 \pm 0.2$ | $30.8 \pm 1$ | $6.5 \pm 0.1$ | $31.4 \pm 0.8$ | $7.3 \pm 0.2$ | $29 \pm 1.1$ | $10.3 \pm 1$ | $\mathbf{14.6 \pm 1.5}$ | $\mathbf{49 \pm 1.9}$ |

Table 5: Ablation studies on different attack methods. We compared our SHIFT attack with PGD, MinBest and PA-AD with 1/255,3/255,15/255 budgets and report reward and deviation rate.

| Freeway | PGD-1/255 | | PGD-3/255 | | PGD-15/255 | | MinBest-1/255 | | MinBest-3/255 | |
|---|---|---|---|---|---|---|---|---|---|---|
| | Recons. | Wass.($\times 10^{-3}$) | Recons. | Wass.($\times 10^{-3}$) | Recons. | Wass.($\times 10^{-3}$) | Recons. | Wass.($\times 10^{-3}$) | Recons. | Wass.($\times 10^{-3}$) |
| DP-DQN | $3.45 \pm 0.3$ | $3.1 \pm 0.2$ | $3.50 \pm 0.3$ | $7.4 \pm 0.3$ | $4.36 \pm 0.29$ | $31 \pm 1$ | $3.45 \pm 0.3$ | $3.7 \pm 0.2$ | $3.53 \pm 0.3$ | $9 \pm 0.4$ |
| | MinBest-15/255 | | PA-AD-1/255 | | PA-AD-3/255 | | PA-AD-15/255 | | Ours | |
| | Recons. | Wass.($\times 10^{-3}$) | Recons. | Wass.($\times 10^{-3}$) | Recons. | Wass.($\times 10^{-3}$) | Recons. | Wass.($\times 10^{-3}$) | Recons. | Wass.($\times 10^{-3}$) |
| DP-DQN | $5.35 \pm 0.2$ | $40 \pm 1$ | $3.47 \pm 0.29$ | $4.5 \pm 0.2$ | $3.60 \pm 0.29$ | $12 \pm 0.1$ | $6.06 \pm 0.18$ | $55 \pm 0.2$ | $\mathbf{1.02 \pm 0.5}$ | $\mathbf{1.1 \pm 0.2}$ |

Table 6: Average reconstruction error and Wasserstein distance of states from a randomly sampled episode under various attack scenarios. Wasserstein-1 distance is calculated between the current perturbed state and the previous step's true state, scaled by 1,000 for readability.

| Pong | PGD (Temporally Coupled) |
|---|---|
| | Reward |
| DQN | $-21 \pm 0$ |
| SA-DQN | $-21 \pm 0$ |
| DP-DQN | $20 \pm 1.73$ |

Table 7: Temporally Coupled PGD attack performance with $\epsilon = 15/255$ and $\bar{\epsilon} = 7.5/255$

| Attack | Defense | Pong Reward | Freeway Reward |
|---|---|---|---|
| **B&C** | DQN | $-21 \pm 0.00$ | $23 \pm 0.00$ |
| | SA-DQN | $11 \pm 0.00$ | $25 \pm 0.00$ |
| | Diffusion History | $20 \pm 1.41$ | $27.2 \pm 0.68$ |
| **Blurred Observations** | DQN | $-21 \pm 0.00$ | $18 \pm 0.00$ |
| | SA-DQN | $-20 \pm 0.00$ | $27 \pm 0.00$ |
| | Diffusion History | $20 \pm 0.58$ | $33.2 \pm 0.37$ |
| **Rotation Degree 1** | DQN | $-20 \pm 0.00$ | $26.6 \pm 0.45$ |
| | SA-DQN | $-18 \pm 0.00$ | $21 \pm 0.00$ |
| | Diffusion History | $14.6 \pm 2.68$ | $27.6 \pm 0.45$ |
| **Shifting (1,0)** | DQN | $-21 \pm 0.00$ | $26 \pm 0.00$ |
| | SA-DQN | $-21 \pm 0.00$ | $24 \pm 0.00$ |
| | Diffusion History | $17.8 \pm 2.85$ | $27.2 \pm 0.37$ |

Table 8: Performance of high-sensitivity direction attacks in (Korkmaz, 2023).

| Pong | Defense | Reward |
|---|---|---|
| **Rotation Degree 3** | Diffusion History | $20 \pm 0.71$ |
| **Shifting (2,1)** | Diffusion History | $18.8 \pm 1.79$ |

Table 9: Large scale rotation and shift attacks against fine-tuned diffusion based defense

**Comparison with Previous Attack Methods.** We provide a complete comparison between PGD (Zhang et al., 2020a), MinBest (Huang et al., 2017), PA-AD (Sun et al., 2021) with budget 1/255,3/255,15/255 and our attack methods in Table 5, which reports both the rewards and the deviation rate of each method. Note that PGD, MinBest and PA-AD do not have the target action, thus we can only compare the deviation rate. The results in Table 5 show our attack method achieves the best attack performance against both SA-DQN and DP-DQN in terms of both reward and deviation rate. Furthermore, we compare the average reconstruction loss of perturbed states and the average Wasserstein-1 distance between a perturbed state and the previous step's true state across a randomly

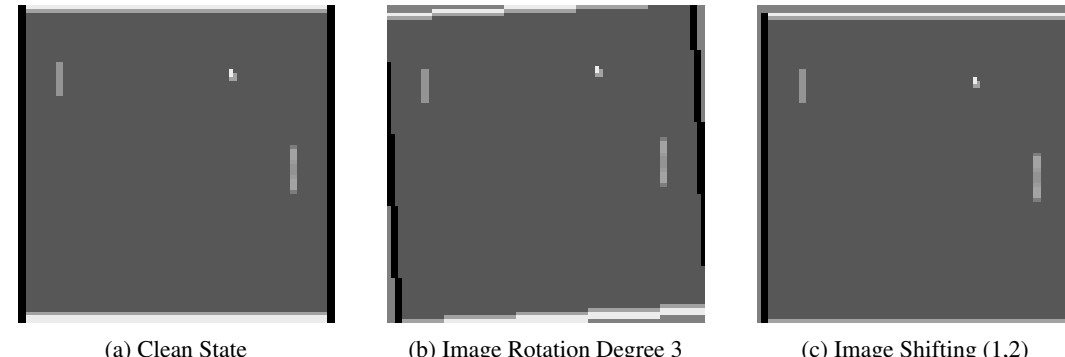

| (a) Clean State | (b) Image Rotation Degree 3 | (c) Image Shifting (1,2) |

Figure 6: Perceptual impact of large scale rotation and shifting on Atari Pong.

| Freeway | **Wasserstein Distance** |
|---|---|
| **B&C** | $0.036 \pm 0.004$ |
| **Blurred Observations** | $0.006 \pm 0.003$ |
| **Rotation Degree 1** | $0.006 \pm 0.004$ |
| **Shifting (1,0)** | $0.07 \pm 0.001$ |
| **Ours** | $0.001 \pm 0.0002$ |

Table 10: Average Wasserstein distance of the high-sensitivity direction attacks in Korkmaz (2023) across a randomly sampled episode. The Wasserstein distance is the Wassersterin-1 distance calculated between the current perturbed state and the previous step's true state.

| Pong | DDPM | | | EDM | | |
|---|---|---|---|---|---|---|
| | Reward | Manipulation Rate(%) | Deviation Rate(%) | Reward | Manipulation Rate(%) | Deviation Rate(%) |
| **DQN** | $-20.6 \pm 0.5$ | $76.6 \pm 1$ | $83.6 \pm 1$ | $-20.7 \pm 0.5$ | $87.1 \pm 1.9$ | $89.6 \pm 1.7$ |
| **Diffusion History** | $5.4 \pm 5.6$ | $15.1 \pm 0.4$ | $45.2 \pm 0.3$ | $6.0 \pm 6.2$ | $8.4 \pm 0.5$ | $25.3 \pm 0.9$ |
| **Sampling Time** | $\sim$5 sec | | | $\sim$0.2 sec | | |

Table 11: Ablation studies on EDM and DDPM diffusion architectures.

sampled episode in Table 6. The Wasserstein distance was proposed in Wong et al. (2019) as an alternative perturbation metric to $l_p$ distances, which measures the cost of moving pixel mass and can represent image manipulations more naturally than the $l_p$ distance. We argue that the reconstruction error captures static stealthiness of state perturbation, while the Wasserstein distance to the previous state captures dynamic stealthiness. Both metrics measure the stealthiness of an attack method and lower the value means better stealthiness. Our attack method achieves the best stealthiness according to Table 6. The superb attack performance and stealthiness brought by our method justifies the use of the conditional diffusion model to generate attacks.

**Temporally Coupled PGD Attack in Atari Environments.** We have implemented a PGD version of the temporally coupled attack introduced in Liang et al. (2024) in Atari environments and tested it against SA-DQN and DP-DQN. The results are in Table 7. The results show that the temporally coupled PGD attack with $\epsilon = 15/255$ and $\bar{\epsilon} = 7.5/255$ could compromise SA-DQN but not diffusion based defense DP-DQN, which indicates the challenge of adapting this attack to Atari environments with raw-pixel input.

**Performance and stealthiness of the high-sensitivity direction attacks in Korkmaz (2023).** Korkmaz (2023) proposes various high-sensitivity direction based attacks that can generate perturbed states that are visually imperceptible and semantically different from the clean states, including changing brightness and contrast(B&C), image blurring, image rotation and image shifting. These attack methods reveal the brittleness of robust RL methods such as SA-DQN, but they mainly target changes in visually significant but non-essential semantics. For example, the relative distance between the pong ball and the pad will remain the same after brightness and contrast changes or image shifting in the Pong environment. Consequently, the perturbed images generated by these methods can potentially be purified by a diffusion model. To confirm this, we have conducted new experiments, showing that (1) the Diffusion History defense with a diffusion model trained from clean data only is able to defend against BC, blurring, and small scale rotation and shifting attacks (see Table 8),

and (2) when the diffusion model is fine-tuned by randomly applying image rotations or shifting during training, the Diffusion History defense can mitigate large scale image rotations and shifting considered in Korkmaz (2023) (see Table 9). In contrast, our diffusion guided attack can change the decision-relevant semantics of the images, such as moving the Pong ball to a different position without changing other elements in the Pong environment as shown in Figure 1). This is the key reason why our attack can bypass strong diffusion-based defense methods.

Furthermore, Korkmaz (2023) claims their attacks are imperceptible by comparing the perturbed state $\tilde{s}_t$ and the true state $s_t$. However, we found that this only holds for small perturbations. For example, the Rotation attack with degree 3 and Shifting attack (1,2) in the Pong environment considered in their paper can be easily detected by humans (see Figure 6). Further, their metric for stealthiness is static and does not consider the sequential decision-making nature of RL. In contrast, our attack method aims to stay close to the set of true states $S^*$ to maintain static stealthiness (Definitions 1 and 2) and align with the history to achieve dynamic stealthiness (Definitions 4 and 5). These are novel definitions for characterizing stealthiness in the RL context. The static stealthiness is demonstrated through the low reconstruction loss of our method shown in Figure 3a. We further compare the Wasserstein distance between a perturbed state and the previous step's true state as a metric to measure stealthiness in Table 10. The results show that the perturbed states generated by our diffusion-based attack stay much closer to the previous step's true states in terms of Wasserstein distance compared with the various attack methods in Korkmaz (2023).

**Ablation on DDPM and EDM diffusion architectures.** We compare DDPM and EDM in terms of attack efficiency and computational cost in Table 11. The results show that EDM and DDPM exhibit similar attack performance. However, DDPM is significantly slower than EDM in terms of sampling time (the average time needed to generate a single perturbed state during testing), making DDPM incapable of generating real-time attacks during testing. This validates the selection of EDM as the diffusion model architecture for constructing our attacks.

