# OpenReview forum: "Diffusion Guided Adversarial State Perturbations in Reinforcement Learning"
_ICLR.cc/2025/Conference — Submitted to ICLR 2025_

### Official Review · Reviewer_D5uT · 2024-11-02

**Soundness:** 3
**Presentation:** 3
**Contribution:** 3
**Rating:** 6
**Confidence:** 3

**Summary:**

This paper proposes a novel non-$\ell_p$ attack algorihtm for image-observation reinforcement learning, based on a history-conditioned diffusion model. The generated attacks are semantically meaningful while misleading to the agent. Experiments show that the proposed SHIFT attack can significantly break existing robust RL learning algorithms that are mainly designed for $\ell_p$ attacks.

**Strengths:**

- The paper points out the limitation of mostly-studied $\ell_p$ attack model for image-observation reinforcement learning environments. By utilizing a denoising diffusion probabilities model (DDPM), the paper achieves stealthy and realistic attack by altering the semantic meaning of the original state.
- The paper clearly defines the concept of valid and realistic states and adopts an autoencoder to enhance the realism of the generated attacks.
- Comparison with existing methods show that SHIFT can lower the reward of agent while having low state reconstruction error.
- The paper also proposes a possible defense method agains the new attack model.

**Weaknesses:**

- Although the proposed attack uses methods such as autoencoder guidance to enhance the realism of the perturbed states, it is not guaranted or bounded like $\ell_p$ attacks, making it hard to compare and evaluate the stealthiness of the perturbations. It is not clear to me whether the reconstruction error can effectively represent the realism of the state perturbation. It would be better if the authors can provide a gif or video showing the full perturbed trajectory.
- The experiments are not very informative. It is not surprising that RL agents learned via $\ell_p$ attack assumptions will break under the proposed attack. But more empirical study can be done to verify the effectiveness of the proposed design. For example, how does the varying attack strengh influence the attack effects?

**Questions:**

As the author mentioned, the proposed method uses a myopic target action manipulation objective which can be sub-optimal. Is there a way to improve it? For example, how can it be combined with RL-based attack methods such as PA-AD?

---

> ### Author Response · Authors · 2024-11-21
> **Author Responses**
>
> Dear Reviewer D5uT:
>
> Thank you for your helpful and insightful feedback. We will address your concerns point by point in the following.
>
> **Q1: Can the reconstruction error effectively represent the realism of state perturbation?**
>
> A1: Thank you for this important comment. As suggested by the reviewer, we have created two gifs, one showing the perturbed trajectory under our attacks constrained by the realism detection, and one showing the unperturbed trajectory for comparison and uploaded them in the updated supplementary material.
>
> We have further explored the Wasserstein-1 distance between a perturbed state and the true state of the previous time step as another realism metric shown in **Table 2-2** in the general response. As argued in [1], the Wasserstein distance captures the cost of moving pixel mass and represents image manipulation more naturally than the $l_p$ distance. Thus a small Wasserstein distance from previous true states shows that even when the agent is aware of the true previous state $s_{t-1}$, the perturbed state $\tilde{s}_t$ generated by our attack is more stealthy than other attacks. The results shown in **Table 2-2** state that our attack method achieves both the lowest Wasserstein distance and the lowest reconstruction error. This experiment further proves that our attack method can generate realistic attacks and achieve superb attack performance at the same time.
>
> **Q2: More empirical studies are needed such as ablation on varying attack strength.**
>
> A2: We included the ablation study on varying guidance strength in Appendix E in the original submission, where we also included ablation studies on effectiveness of the realism guidance and target action selection methods. We agree with the reviewer that more evaluation results would help prove our attack method’s effectiveness and soundness. Thus, we have conducted new experiments including (1) an experiment for the Atari Roadrunner environment (see **Table 1** in the general response), (2) an experiment that compares different attack methods including the newly added PA-AD attack (see **Table 2** in the general response), and (3) an experiment that compares the DDPM and EDM diffusion methods (see **Table 5** in the general response). We hope these additional ablation studies can address your concern.
>
> **Q3: Use PA-AD or other non-myopic target action selection methods.**
>
> A3: We would like to thank the reviewer for this insightful comment. Combining PA-AD or other non-myopic target action selection methods with our conditional diffusion based attack is indeed a promising direction. A simple strategy is to adopt a two-step approach, similar to what we did in our paper, where one first applies a non-myopic target action selection method to select target actions with some long-term attack objective in mind, and then uses our diffusion-based attack to approximate the goal. An important challenge, however, is that a diffusion-based approach cannot guarantee that the target actions are alway chosen both due to the randomness of the diffusion model and the requirement for maintaining realism and history alignment. Therefore, to provide a performance guarantee, one needs to measure the expected success rate of a conditional diffusion model for a given set of target actions, which are then optimized toward some long-term goal that is achievable. We believe this requires a non-trivial extension of our approach and is an interesting future direction to explore.
>
> We sincerely hope our responses and additional experiment results have addressed all your concerns. If so, we kindly hope that you consider increasing the rating of our paper. Please do not hesitate to add further comments if you have any additional questions or need further clarifications.
>
> [1] Eric Wong, Frank R. Schmidt, and J. Zico Kolter. Wasserstein Adversarial Examples via Projected Sinkhorn Iterations. ICML 2019.

---

> > ### Author Response · Authors · 2024-12-02
> > **Did our responses address all your concerns?**
> >
> > Dear Reviewer D5uT,
> >
> > Thank you once again for your thoughtful feedback. As the rebuttal period approaches its conclusion, we hope to hear whether our responses address your concerns. Below is a summary of our rebuttal:
> > 1. **Realism Metric**: We provided both original and perturbed trajectory GIFs in the supplementary material and introduced the Wasserstein-1 distance as an additional realism metric, demonstrating that our attack achieves both static and dynamic stealthiness.
> > 2. **More Empirical Experiments**: We added new experiments, including Roadrunner evaluations, comparisons with PA-AD, and analysis of DDPM versus EDM diffusion methods, which reinforce our method's effectiveness.
> > 3. **Non-Myopic Action Selection**: We discussed integrating PA-AD and non-myopic action selection methods with our approach, identifying challenges and potential future directions.
> >
> > We hope these updates address your concerns and welcome your further feedback.

---

### Official Review · Reviewer_37tk · 2024-11-03

**Soundness:** 1
**Presentation:** 1
**Contribution:** 1
**Rating:** 3
**Confidence:** 4

**Summary:**

The submission claims to find that the effectiveness of the current defenses is due to a fundamental weakness of the existing $\ell_p$-norm constrained attacks. Furthermore, the submission proposes a method to go beyond the $\ell_p$-norm bounded adversarial attacks in deep reinforcement learning. The submission evaluates its proposed attacks in Atari games and argues that the proposed attack method of the submission lowers the cumulative rewards of the agent by 50%.

**Strengths:**

AI safety and robustness is an important research area.

**Weaknesses:**

The major claimed contributions of the submission have been previously both mentioned and analyzed in previous work [1]. However, the submission does not refer to these studies, and furthermore, within the existing prior work the main claimed contributions of the submission are rather misplaced and inaccurate. The paper [1] already extensively studies and demonstrates that both deep reinforcement learning policies and current defenses, i.e. robust deep reinforcement learning, are not robust against semantically meaningful adversarial attacks and this study further reveals the need to have robustness beyond $\ell_p$-norm bounded attacks.

Not only has the necessity of considering beyond  $\ell_p$-norm bounded attacks already been discussed in previous work, furthermore the approach proposed in this paper [1] achieves higher degradation on the policy performance without even having access to the training details, the policy network (i.e. black-box adversarial attacks), and further without even training any additional network to produce such adversarial examples.

The submission substantially lacks appropriate references, and further positioning itself within the existing prior work and clarifying its main contributions within these studies. The claimed contributions of the submission are misplaced and incorrect.

[1] Adversarial Robust Deep Reinforcement Learning Requires Redefining Robustness. AAAI Conference on Artificial Intelligence, AAAI 2023.

Furthermore, the submission lacks main technical details to interpret their experimental results. Not a single experimental detail is provided regarding deep reinforcement learning. These details are essential for reproducibility and further to interpret and analyze the experimental results provided in the submission. However, the submission does not provide any information on this.

The submission only tests their algorithm in 3 games from Atari. However, in adversarial deep reinforcement learning it is usually tested in more games [1,2,3,4]. In particular, RoadRunner is missing from the baseline comparison.

[1] Robust deep reinforcement learning against adversarial perturbations on state observations, NeurIPS 2020.

[2] Robust Deep Reinforcement Learning through Adversarial Loss, NeurIPS 2021.

[3] Adversarial Robust Deep Reinforcement Learning Requires Redefining Robustness, AAAI 2023.

[4] Detecting Adversarial Directions in Deep Reinforcement Learning to Make Robust Decisions, ICML 2023.

The submission also refers to main concepts in the adversarial machine learning literature with inaccurate wording. For instance, in the introduction the submission writes:

*“by poisoning its observation (Huang et al., 2017; Zhang et al., 2020a)”*

However, poisoning attacks in adversarial machine learning literature refer to completely something else and these papers are not poisoning attacks. These papers are test time attacks. Thus, it is misleading to use the word poisoning here.

One thing I find ineffective is that the submission refers to a long list of papers such as these [1,2,3,4,5], however, somehow still misses the prior work that substantially coincides with the main claimed contributions of the submission and even further these prior studies already demonstrate the claimed contributions of this submission.

[1] Kangjie Chen, Shangwei Guo, Tianwei Zhang, Xiaofei Xie, and Yang Liu. Stealing deep reinforcement learning models for fun and profit. ACM Asia Conference on Computer and Communications Security, 2021.

[2] Mengdi Huai, Jianhui Sun, Renqin Cai, Liuyi Yao, and Aidong Zhang. Malicious attacks against deep reinforcement learning interpretations. ACM SIGKDD International Conference on Knowledge Discovery & Data Mining, 2020.

[3] Yunhan Huang and Quanyan Zhu. Deceptive reinforcement learning under adversarial manipulations on cost signals. Decision and Game Theory for Security (GameSec), 2019.

[4] Zikang Xiong, Joe Eappen, He Zhu, and Suresh Jagannathan. Defending observation attacks in deep reinforcement learning via detection and denoising. Machine Learning and Knowledge Discovery in Databases: European Conference 2023.

[5] Inaam Ilahi, Muhammad Usama, Junaid Qadir, Muhammad Umar Janjua, Ala I. Al-Fuqaha, Dinh Thai Hoang, and Dusit Niyato. Challenges and countermeasures for adversarial attacks on deep reinforcement learning. ArXiv 2020.

**Questions:**

See above

---

> ### Author Response · Authors · 2024-11-21
> **Author Responses(1)**
>
> Dear Reviewer 37tk:
>
> Thank you for your insightful feedback. We will address all your concerns point by point in the following.
>
> **Q1: The main claimed contributions have been mentioned and analyzed in [1].**
>
> A1: We would like to thank the reviewer for pointing us to this important study [1]. We have updated the introduction section of our paper to acknowledge the contribution of [1] and also included a detailed comparison with [1] in the related work section and the evaluation section. However, after carefully comparing our work with [1] and conducting additional environments (as the code in [1] is not publicly available, we have tried to reproduce some of their results in our environment), we found that although both [1] and our work consider attacks beyond $l_p$ norm constraint, the two attack methods are significantly different in multiple aspects and our paper has made substantial new contributions beyond what is already considered in [1].
>
> First, our attack is able to change the **essential semantics that matter for decision making, making it much harder to defend.** [1] shows that by following high-sensitivity directions, including changing brightness and contrast, image blurring, image rotation and image shifting, it is possible to generate perturbations that are visually imperceptible and semantically different from the original state. These attack methods reveal the brittleness of robust RL methods such as SA-DQN, but they **mainly target changes in visually significant but non-essential semantics.** For example, the relative distance between the pong ball and the pad will remain the same after brightness and contrast changes or image shifting in the Pong environment. Consequently, the perturbed images generated by these methods can potentially be denoised by a diffusion model. To confirm this, we have conducted new experiments, showing that (1) the Diffusion History defense with a diffusion model trained from clean data only is able to defend against B&C, blurring, and small scale rotation and shifting attacks (see **Table 4-1** in the general response), and (2) when the diffusion model is fine-tuned by randomly applying image rotations or shifting during training, the Diffusion History defense can mitigate large scale image rotations and shifting considered in their paper (see **Table 4-2** in the general response). In contrast, our diffusion guided attack can **change the decision-relevant semantics of images**, such as moving the Pong ball to a different position without changing other elements in the Pong environment as shown in Figure 1 e) in the paper. This is the key reason why our attack can bypass strong diffusion based defense methods.
>
> Second, **our attack is stealthy from both static and dynamic perspectives.** [1] claims that the perturbed states generated by their high-sensitivity direction based attacks are imperceptible by comparing the perturbed state $\tilde{s_t}$ and the true state $s_t$. However, we found that this only holds for small perturbations. For example, the Rotation attack with degree 3 and Shifting attack (1,2) in the Pong environment considered in their paper can be easily detected by humans (see Figure 6 in Appendix E in our revised paper). Further, their metric for stealthiness is static and does not consider the sequential decision-making nature of RL. In contrast, our attack method aims to stay close to the set of true states $S^*$ to maintain **static stealthiness** (Definitions 1 and 2 in our paper) and align with the history to achieve **dynamic stealthiness** (Definitions 4 and 5). These are novel definitions for characterizing stealthiness in the RL context. The static stealthiness is demonstrated through the low reconstruction loss of the perturbed states generated by our method shown in Figure 3 a) in our paper. To better illustrate that our attacks are stealthy from a dynamic perspective, we have added an ablation study to compare the average Wasserstein-1 distance between a perturbed state and the previous step’s true state across a randomly sampled episode (see **Tables 2-2 and 4-3** in the general response). As argued in [3], the Wasserstein distance captures the cost of moving pixel mass and represents image manipulation more naturally than the $l_p$ distance. The results show that even when the agent is aware of the true previous state $s_{t-1}$, the perturbed state $\tilde{s}_t$ generated by our attack is more stealthy than other attacks including the attack methods in [1] (**Table 4-3** in the general response).
>
> (Continue in the next comment)

---

> ### Author Response · Authors · 2024-11-21
> **Author Responses(2)**
>
> In conclusion, although both our attack method and [1] attempt to go beyond $l_p$ norm bounded attacks, the attack methods in [1] focus on policy independent attacks that are visually imperceptible and can compromise defenses like SA-DQN but these attacks cannot change the essential semantics of image observations and fail to fight against diffusion-based defenses. In contrast, our attack method used a conditional diffusion model to generate perturbed states that can change the essential semantics of the image data while remaining stealthy to bypass the strong defenses including diffusion-based defenses.
>
> **Q2: The submission substantially lacks appropriate references. The claimed contributions of the submission are misplaced and incorrect.**
>
> A2: We thank the reviewer for the comment and for pointing out the missing related work. After careful review, we found that seven of the nine papers mentioned by the reviewer were included in our original submission. The remaining two papers [1] [2] both focus on high-sensitivity direction attacks, where the approaches are significantly different from ours as discussed above. We have carefully revised our paper to acknowledge their contributions and highlight the novelty of our work. We have also included a discussion of these papers in the related work section of the revised submission. As the code in [1] and [2] are not publicly available, we were not able to include the detection method in [2] as a baseline, but we have managed to implement the attack methods in [1] and showed that they cannot compromise diffusion-based defenses as discussed above.
>
> **Q3: Lack of technical and experiment details for reproducibility.**
>
> A3: We discussed our experiment setting in Appendix D.6 in the original submission, where we reported the hyperparameters for training the conditional diffusion model and the guidance strength during testing. As we mentioned there, we used the default parameters setting in their original papers for other defense methods. We also provided the source code together with our paper during submission. In response to the reviewer’s concern, we have added a section on pre-processing the Atari environments for better reproducibility. In the revision, we have provided a set of new ablation study results, as discussed in the general response. For each of them, we have provided some insights into the results and connected them with our approach in Section 3. For example, we have provided a detailed comparison of our attack with the high-sensitivity direction attacks in [1] and highlighted the importance of considering both static and dynamic stealthiness in deep reinforcement learning with image input. We have also adapted the Wasserstein distance perturbation metric [3] originally proposed for adversarial examples in deep learning to the deep reinforcement learning context by considering the sequential decision-making nature of RL. We hope these efforts adequately address the reviewer's concern about the lack of technical details.
>
> **Q4: Missing Roadrunner results.**
>
> A4: As suggested by the reviewer, we have added new experiment results on Roadrunner in **Table 1** in the general response.
> We have retrained the vanilla DQN model and the SA-DQN model because the pretrained vanilla DQN and SA-DQN models provided by the SA-MDP paper [4] do not work in the RoadRunner environment. The result shows that similar to other Atari environments we evaluated before, our attack is able to compromise SA-DQN and Diffusion History defenses in the Roadrunner environment.
>
> **Q5: Improper use of the word “poisoning.”**
>
> A5: We carefully reviewed the paper and found that we misused the word “poisoning” once. We thank the reviewer for catching this and have corrected it in the revision.
>
> We sincerely hope our responses and additional experiment results have addressed all your concerns. If so, we kindly hope that you consider increasing the rating of our paper. Please do not hesitate to add further comments if you have any additional questions or need further clarification.
>
> [1] Ezgi Korkmaz, Adversarial Robust Deep Reinforcement Learning Requires Redefining Robustness, AAAI 2023.
>
> [2] Ezgi Korkmaz and Jonah Brown-Cohen, Detecting Adversarial Directions in Deep Reinforcement Learning to Make Robust Decisions, ICML 2023.
>
> [3] Eric Wong, Frank R. Schmidt, and J. Zico Kolter, Wasserstein Adversarial Examples via Projected Sinkhorn Iterations, ICML 2019.
>
> [4] Huan Zhang et al., Robust Deep Reinforcement Learning against Adversarial Perturbations on State Observations, NeurIPS 2020.

---

> > ### Author Response · Authors · 2024-12-02
> > **Did our responses address all your concerns?**
> >
> > Dear Reviewer 37tk,
> >
> > Thank you for your thoughtful feedback. As the rebuttal period approaches its conclusion, we hope to hear whether our responses address your concerns. Below is a brief summary of our rebuttal:
> >
> > 1. **Key Contributions Beyond [1]**: While [1] focuses on high-sensitivity directions (e.g., brightness, contrast, rotation), these attacks fail against diffusion-based defenses as they do not alter decision-relevant semantics. Our method leverages a conditional diffusion model to modify essential semantics (e.g., Pong ball position) while maintaining static and dynamic stealthiness, as demonstrated by low reconstruction loss and minimal Wasserstein distance.
> > 2. **References and Novelty**: We acknowledged [1] and [2] in the revised submission, highlighting their contributions and distinctions from our approach. Despite the lack of code, we implemented the four attacks in [1] and showed their ineffectiveness against diffusion-based defenses.
> > 3. **Technical Details and Reproducibility**: We detailed our experimental setup, hyperparameters, and Atari environment preprocessing in the revised submission. Additionally, we included new ablation studies to emphasize the importance of static and dynamic stealthiness in our method.
> > 4. **Roadrunner Results**: We extended our experiments to the Roadrunner environment, demonstrating that our attack compromises both SA-DQN and diffusion-based defenses, consistent with results from other environments.
> >
> > We hope these clarifications address your concerns and welcome any further feedback or questions.

---

> > > ### Comment · Reviewer_37tk · 2024-12-02
> > >
> > > I thank the authors for their response. However, there are several incorrect statements made in the author's response that I must address.
> > >
> > > *”Authors: First, our attack is able to change the essential semantics that matter for decision making, making it much harder to defend. [1] shows that by following high-sensitivity directions, including changing brightness and contrast, image blurring, image rotation and image shifting, it is possible to generate perturbations that are visually imperceptible and semantically different from the original state. These attack methods reveal the brittleness of robust RL methods such as SA-DQN, but they mainly target changes in visually significant but non-essential semantics. For example, the relative distance between the pong ball and the pad will remain the same after brightness and contrast changes or image shifting in the Pong environment.”*
> > >
> > > This is an incorrect statement. The several methods introduced in the paper [1] indeed change the essential semantics of the environment. In particular, perspective transform and rotation indeed change the distance of the pong ball and the pad.
> > >
> > > Authors currently still missing quite critical issues that are essential to adversarial machine learning. The proposed methods in [1] decrease the policy performance of deep reinforcement learning policies by around 90% without even having access to the policy details, i.e. network, algorithm, training details or even the training environment. Thus, an adversarial attack that does not require any training or any access to the victim policy’s private information is a more dangerous and powerful attack.
> > >
> > > Furthermore, the submission still positions itself as the first paper that goes beyond $\ell_p$-norm attacks. This is also incorrect. Furthermore, there are currently studies that achieve 90% damage on the policy performance with no additional training or having access to the history of the policy or any training details of the policy.
> > >
> > > The submission still positions itself as a paper that shows adversarial training, i.e. robust deep reinforcement learning, does not work against beyond $\ell_p$-norm bounded attacks. This is also incorrect. This is already known. Prior studies already demonstrated this.
> > >
> > > I will keep my score.

---

> ### Author Response · Authors · 2024-12-02
>
> Dear Reviewer 37tk:
>
> Thanks for your feedback on our rebuttal and we will provide further clarifications on your concerns.
>
> **Q1**: [1]'s methods can change the essential semantics and decrease 90% of the policy's performance without accessing the victim's policy.
>
> **A1**: We acknowledge that rotations and perspective transformations in [1] may alter the absolute distances, such as between the Pong ball and the paddle. However, it is crucial to note that these changes do not affect the semantic meaning based on our **Definition 3**: after projecting perturbed states onto the true state set, the projected states remain close to the original states. This explains why diffusion-based defenses effectively counter the attacks in [1], as shown in Tables 4-1 and 4-2 in the general rebuttal, by recovering true states from perturbed ones.
>
> Additionally, we note that large-scale transformations in [1] are not stealthy. For example, we provide images of 3-degree rotations in Appendix E, where the perturbed states are visually distinguishable from true states, violating static stealthiness.
>
> **Q2**: Authors currently still missing quite critical issues that are essential to adversarial machine learning.
>
> **A2**: We agree that the policy-independent and history-ignorant attack given in [1] is lightweight to implement and opens up an interesting direction. However, it also has fundamental limitations. First, it cannot bypass diffusion-based adversarial training and may generate unrealistic perturbations, as discussed above and shown in the revised paper. Second and more importantly, it largely ignores the sequential decision-making nature of RL. All the attacks implemented in [1] are applied to individual states in a myopic way. Thus, they either do not change the semantics essential to decision-making (e.g., when the same rotation operation is applied to all the states as in the paper) or cannot generate history-consistent perturbations (e.g., when different operations are applied in each state). Our approach aims to address these limitations.
>
> **Q3**: Our work is the first to claim an attack beyond $l_p$-norm.
>
> **A3**: We would like to clarify that we did not claim to be the first to propose attacks beyond $l_p$-norm in our paper. Furthermore, we have cited [1] in our revised manuscript and explicitly identified it as a beyond $l_p$-norm attack in the introduction, related work, and evaluation sections. We will further clarify this by incorporating the discussion above into the revision.
>
>  **Q4**: The statement "adversarial training is not effective against beyond $l_p$-norm attacks" is proved in previous work.
>
> **A4**: We respectfully disagree with the reviewer on this. Before our work, it was unclear if there were any adversarial perturbation attacks, including those beyond $l_p$-norm constraints, that could bypass all existing robust RL approaches while remaining stealthy. As shown in Tables 4-1 and 4-2, a diffusion model trained with adversarial data generated by the attacks in [1] can successfully defend against these beyond $l_p$-norm attacks in [1]. This demonstrates that adversarial training can be effective against such attacks.
>
> Furthermore, we only claim that our proposed attack compromises existing robust RL methods. We do not rule out the possibility that novel adversarial training-based defenses could effectively counter our attack.
>
> We hope our clarifications address your concerns, and we welcome any further feedback.

---

### Official Review · Reviewer_L64s · 2024-11-03

**Soundness:** 2
**Presentation:** 2
**Contribution:** 2
**Rating:** 5
**Confidence:** 4

**Summary:**

This paper introduces SHIFT, a diffusion-based adversarial attack that targets RL agents in vision-based environments by creating realistic, history-aligned state perturbations that go beyond traditional lp-norm attacks. Unlike existing methods, SHIFT generates semantic changes that significantly impair the agent's performance, bypassing even the most advanced defenses. Results demonstrate the attack's efficacy, reducing cumulative rewards by over 50% in Atari games, underscoring the need for more robust defenses in RL.

**Strengths:**

1. This work proposes semantic-level RL attacks using conditional diffusion models that balance semantic changes, realism, and historical consistency. The insight is novel.
2. Identifies a fundamental weakness in lp-norm attacks - their inability to meaningfully alter state semantics despite large perturbation budgets.
3. Employs EDM to enhance generation efficiency, making the approach more feasible

**Weaknesses:**

1. Despite using EDM and weighting joint, the paper lacks any systematic analysis of attack efficiency and computational costs.
2. While reporting larger attack budgets, results are limited to PGD and MinBest baselines, missing broader comparative analysis.
3. Experiments are restricted to only three Atari environments, providing insufficient evidence for the method's generalizability.
4. Overall Soundness: While the core idea is interesting, the paper falls short in rigor - lacking ablation studies, methodology analysis, and comprehensive experiments. The current evaluation scope is not convincing enough to support the claims.

**Questions:**

1. The Manipulation Rate and Deviation Rate metrics appear exclusive to SHIFT's diffusion-based approach, raising questions about fair comparison with non-diffusion methods. The necessity of diffusion models needs stronger justification.
2. The paper lacks crucial comparison between DDPM and EDM in terms of both effectiveness and efficiency. This missing analysis weakens the justification for the chosen architecture.
3. Overlooks recent related work [1] about temporally-coupled perturbations

[1]Game-Theoretic Robust Reinforcement Learning Handles Temporally-Coupled Perturbations, Liang et al, ICLR 2024

---

> ### Author Response · Authors · 2024-11-21
> **Author Responses (1)**
>
> Dear Reviewer L64s:
>
> Thank you for your insightful feedback on our works. We will address your concerns point by point in the following by providing additional results and clarifications.
>
> **Q1: Lack of computational costs analysis of our attack.**
>
> A1: We are delighted to provide detailed computational costs of our attacks here. During the training stage, it takes around 1.5 hours to train both the conditional diffusion model and the autoencoder based realism detector. We remark that these two components can be trained in parallel. During the testing stage, our attack takes around 0.2 seconds to generate a perturbed state, making it feasible for real-time attacks.
>
> **Q2: Lack of comparison between DDPM and EDM diffusion architectures.**
>
> A2: We have provided new results to compare DDPM and EDM in terms of attack efficiency and computational cost in the general response (**Table 5**). The results show that EDM and DDPM exhibit similar attack performance. However, DDPM is significantly slower than EDM in terms of running time (the average time needed to generate a single perturbed state during testing), making DDPM incapable of generating real-time attacks during testing. This validates the selection of EDM as the diffusion model architecture for constructing our attacks.
>
> **Q3: Our methods were only tested on three Atari environments.**
>
> A3: We have included new evaluation results on the RoadRunner environment that is widely used in previous work. Please see **Table 1** in the general response, which shows that our attack obtains superb performance against SA-DQN and Diffusion History defenses.
>
> **Q4: Comparison with other attack baselines is insufficient.**
>
> A4: We have included evaluation results for a new attack baseline, PA-AD [2], which is considered one of the strongest attacks in the literature. A complete comparison between PGD, MinBest, PA-AD and our attack methods is given in **Table 2-1** in the general response, which reports both the reward and the deviation rate of each method. We want to emphasize that while the manipulation rate does not apply to PGD, MinBest and PA-AD, the reward and the deviation rate (the fraction of the chosen actions under perturbed states differ from the actions under the true states across an episode) do apply to all the baselines. The results in **Table 2-1** show that our attack method achieves the best attack performance against both SA-DQN and DP-DQN in terms of both reward and deviation rate.
>
> Furthermore, we have added the Wasserstein-1 distance between a perturbed state and the true state in the previous time step as a new metric to measure the dynamic stealthiness of baseline attacks. As argued in [3], the Wasserstein distance captures the cost of moving pixel mass and represents image manipulation more naturally than the $l_p$ distance. We have reported the reconstruction loss of perturbed states and the Wasserstein distance between a perturbed state and the previous step’s true state in **Table 2-2** in the general response. The former captures the static stealthiness while the later captures the dynamic stealthiness as we further elaborated in the general response. The results show that our attack achieves both the lowest reconstruction error and the lowest Wasserstein distance, indicating our attack method achieves best stealthiness from both static and the dynamic perspectives. The superb attack performance and stealthiness of our method justify the use of the conditional diffusion model to generate attacks.

---

> ### Author Response · Authors · 2024-11-21
> **Author Responses(2)**
>
> **Q5: Lack of discussion on [1]**
>
> A5: We thank the reviewer for pointing us to this important study and we have added it to the related work in the updated version of our paper. However, after carefully reading the paper, we found that the temporally coupled attack and the game theoretic defense method proposed in [1] might not directly apply to our setting due to the following reasons.
>
> First, all the experiments in [1] are conducted in MuJoCo environments, where the state spaces are much smaller compared with Atari environments with image input. Further, their approaches are already computationally expensive (both take more than 20 hours) to train in MuJoCo environments. Thus, directly applying them to image domains can be computationally prohibitive, which points to an interesting research direction for further study. Second, the code for [1] is not publicly available at this time so we cannot easily evaluate their attacks and defenses as baselines in Atari environments.
>
> In response to the reviewer’s concern, we have implemented a PGD version of the temporally coupled attack in Atari environments and tested it against SA-DQN and DP-DQN. The results are in the general response **Table 3**. The results show that the temporally coupled PGD attack with $\epsilon = 15/255$ and $\bar{\epsilon} = 7.5/255$ could compromise SA-DQN but not diffusion based defense DP-DQN even with a large perturbation budget, which indicates the challenge of adapting this attack to Atari environments with raw-pixel input. We conjecture that this is because the attack is still constrained by an $l_p$ norm bound, making it difficult to alter the essential semantics of image input.
>
> We sincerely hope our responses and additional experiment results have addressed all your concerns. If so, we kindly hope that you consider increasing the rating of our paper. Please do not hesitate to add further comments if you have any additional questions or need further clarifications.
>
> [1] Liang et al., Game-Theoretic Robust Reinforcement Learning Handles Temporally-Coupled Perturbations, ICLR 2024
>
> [2] Sun, Y., Zheng, R., Liang, Y., & Huang, F. Who Is the Strongest Enemy? Towards Optimal and Efficient Evasion Attacks in Deep RL. ICLR 2022.
>
> [3] Eric Wong, Frank R. Schmidt, and J. Zico Kolter. Wasserstein Adversarial Examples via Projected Sinkhorn Iterations. ICML 2019.

---

> > ### Comment · Reviewer_L64s · 2024-11-27
> > **Response to authors**
> >
> > Thank you for your detailed response and the additional experiments. The supplementary experiments demonstrate that your proposed method has unique advantages. However, I still believe there are several areas in the manuscript that require further development. The additional experiments presented in the rebuttal should be incorporated into the main text to strengthen your statements and conclusions, as these are the basic standards of an ICLR paper.
> > I am raising my score to 5, and after discussing with other reviewers and the Area Chair, I will determine whether to further increase my score.

---

> ### Author Response · Authors · 2024-11-27
> **Thank you for your feedback**
>
> Dear Reviewer L64s:
>
> Thank you for reviewing our rebuttal and for increasing your rating. In response to your valuable feedback, we have further revised our manuscript to include additional experiments conducted during the rebuttal period. The following changes have been made to the main text:
>
> 1. The **RoadRunner** results have been added to **Table 1**.
> 2. **Figure 3** now includes results for PA-AD, temporally coupled, and high-sensitivity direction based attacks, along with the introduction of **Wasserstein distance** as an additional metric to measure stealthiness.
> 3. We have added the ablation study on **DDPM** and **EDM** diffusion architectures.
>
> We believe that these revisions, along with the extra experiments and discussions, strengthen the paper's statements and conclusions.
>
> We appreciate your continued feedback and hope that the revised manuscript addresses your concerns effectively.

---

### Official Review · Reviewer_eNaA · 2024-11-03

**Soundness:** 2
**Presentation:** 2
**Contribution:** 2
**Rating:** 3
**Confidence:** 3

**Summary:**

The paper studies how to generate state perturbations for reinforcement learning, especially perturbation in an unconstrained way, instead of traditional L_p perturbations. The methods are based on diffusion models to generate states with different semantics. The experiments outperforms some existing baselines.

**Strengths:**

1. The paper studies an interesting question of non-L_p attacks, which is largely neglected by existing literature.
2. The methods can scale to image-input domain

**Weaknesses:**

1. One concern/question is that the goal of the paper is not very concrete. The paper said existing methods cannot change the semantics of the image input while this paper can. However, it is not very clear why the attacker has the motivation to change the semantics? In other words, isn't being stealthy beneficial for the attacker?
2. Some newest/recent defense baselines to my best knowledge [1, 2] are not discussed or compared in experiments. These game-theoretical based defense methods are significantly different from the defense mechanisms discussed in the paper by nature, and more importantly agnostic to the attacker model (which means one only need to change the attacker model to non-L_p accordingly to extend the defense to non-L_p). Therefore, it will be important to evaluate how the attacks performs under such kind of defense strategies.

[1] Game-Theoretic Robust Reinforcement Learning Handles Temporally-Coupled Perturbations ICLR 2024
[2] Beyond Worst-case Attacks: Robust RL with Adaptive Defense via Non-dominated Policies ICLR. 2024

**Questions:**

see weakness

---

> ### Author Response · Authors · 2024-11-21
> **Author Responses**
>
> Dear Reviewer eNaA:
>
> Thank you for your valuable and insightful feedback on our work. We will address your questions and concerns point by point.
>
> **Q1: Motivation of changing the semantics.**
>
> A1: As shown in recent work such as DP-DQN [3] and Diffusion History (inspired by [5]), current $l_p$ norm bounded attacks including PA-AD cannot bypass these diffusion-based defense methods in environments with raw-pixel inputs such as Atari games even with a relatively large perturbation budget (please refer to **Table 2-1** in the general response). We argue that the main reason is that these attacks are not able to change the essential semantics of the image input under a reasonable attack budget, so that a diffusion-based defense can purify the noise injected to gain strong defense performance.
>
> The existence of these strong diffusion-based defenses against $l_p$ norm bounded attacks motivate us to develop new attacks that can change the semantics of states to mislead those defense methods to choose non-optimal actions even after purifying the perturbed states. Our attack still remains stealthy from both a static and a dynamic perspective by utilizing a history conditioned diffusion model with realism guidance. The static stealthiness is demonstrated through the low reconstruction loss of the perturbed states generated by our method shown in Figure 3 a) in our paper. To better illustrate that our attacks are stealthy from a dynamic perspective, we have added an ablation study to compare the Wasserstein-1 distance between a perturbed state and the true state in the previous time step. As shown in the general response **Table 2-2**, our attack method has the lowest average Wasserstein distance among all the attacks. As argued in [4], the Wasserstein distance captures the cost of moving pixel mass and represents image manipulation more naturally than the $l_p$ distance. The result shows that even when the agent is aware of the true previous state $s_{t-1}$, the perturbed state $\tilde{s}_t$ generated by our attack is more stealthy than other attacks.
>
> **Q2: Recent defense baselines in [1] and [2].**
>
> A2: We thank the reviewer for pointing us to these studies. We have included them in the related work section of the revised submission. However, after reading both papers carefully, we found that the defense methods proposed in [1] and [2] might not directly apply to our setting due to the following reasons. First, both studies conduct their experiments in MuJoCo environments where the state spaces are much smaller compared with Atari environments with image input. Further, these approaches are already computationally expensive (both take more than 20 hours) to train in MuJoCo environments. Thus, directly applying them to image domains can be computationally prohibitive, which points to an interesting research direction for further study. Second, the code for [1] is not publicly available at this time and the code for [2] only implements MuJoCo environments so we cannot easily evaluate these two defense methods against our attacks in Atari environments.
>
> Although we were not able to implement the game theoretic defense in [1] due to the lack of code and the expected high computational overhead in Atari environments, we would like to point out that the DP-DQN [3] defense currently considered in the paper also adopts a game-theoretic approach by identifying an approximate Stackelberg equilibrium. While the vanilla DP-DQN uses PGD attack to simulate worst case attacks, we have retrained DP-DQN by replacing the bounded PGD attack with our unbounded diffusion guided attack. We trained this modified DP-DQN on the Pong environment with 1 million steps but the reward remained at -21 (the worst case) all the time. This gives evidence that even game theoretical defenses might not be strong enough to defend against our attacks.
>
> We sincerely hope our responses have addressed all your concerns. If so, we kindly hope that you consider increasing the rating of our paper. Please do not hesitate to add further comments  if you have any additional questions or need further clarifications.
>
> [1] Liang et al., Game-Theoretic Robust Reinforcement Learning Handles Temporally-Coupled Perturbations. ICLR 2024
>
> [2] Liu et al., Beyond Worst-case Attacks: Robust RL with Adaptive Defense via Non-dominated Policies. ICLR 2024
>
> [3] Xiaolin Sun and Zizhan Zheng, Belief-Enriched Pessimistic Q-Learning against Adversarial State Perturbations. ICLR 2024
>
> [4] Eric Wong, Frank R. Schmidt, and J. Zico Kolter. Wasserstein Adversarial Examples via Projected Sinkhorn Iterations. ICML 2019.
>
> [5] Zhihe Yang and Yunjian Xu, DMBP: Diffusion model-based predictor for robust offline reinforcement learning against state observation perturbations. ICLR 2024

---

> > ### Author Response · Authors · 2024-12-02
> > **Did our responses address all your concerns?**
> >
> > Dear Reviewer eNaA,
> >
> > Thank you once again for your valuable feedback. As the rebuttal period comes to a close, we hope to hear from you regarding whether our responses have satisfactorily addressed your concerns. Below is a brief summary of our rebuttal:
> > 1. **Motivation for Semantic Attacks**: $l_p$ norm-bounded attacks fail against diffusion-based defenses in raw-pixel environments like Atari. To address this, our attack changes state semantics while maintaining static and dynamic stealthiness, as demonstrated by low reconstruction loss (Fig. 3a) and minimal Wasserstein distance (Table 2-2).
> > 2. **Recent Defense Baselines**: We appreciate the references to [1] and [2]. However, these methods are not directly applicable to Atari environments due to their computational cost and the lack of code. Instead, we tested a game-theoretic approach by retraining DP-DQN with our attack, which consistently performed poorly, underscoring the strength of our method.
> >
> > We hope these clarifications resolve any remaining concerns, and we would greatly appreciate further comments or feedback if needed.

---

### Author Response · Authors · 2024-11-21
**General Response(1)**

We thank all the reviewers for their insightful feedback and constructive criticism regarding our insufficient experiment results. In this general response, we provide more experiment results to address these concerns.

**Table 1: RoadRunner Results under Our Attack**
| RoadRunner | Reward | Manipulation | Deviation |
|:---:|:---:|:---:|:---:|
| No Attack | 13500(0) | NA | NA |
| DQN | 0(0) | 52%(2%) | 70%(3%) |
| SA-DQN | 260(215.41) | 34%(2%) | 54%(1%) |
| Diffusion History | 1480(788.42) | 9%(2%) | 43%(2%) |

We have added new experiment results on Atari RoadRunner as suggested by Reviewers 37tk and L64s. We have retrained the vanilla DQN model and the SA-DQN model because the pretrained vanilla DQN and SA-DQN models provided by the SA-MDP paper do not work in the RoadRunner environment. Our attack obtains superb performance against SA-DQN and Diffusion History defenses.

**Table 2-1: Attack Performance of Different Attack Methods**
| **Freeway** | PGD-1/255 |  | PGD-3/255 |  | PGD-15/255 |  | Minbest-1/255 |  | Minbest-3/255 |  | Minbest-15/255 |  | PA-AD-1/255 |  | PA-AD-3/255 |  | PA-AD-15/255 |  | Ours |  |
|---|---|---|---|---|---|---|---|---|---|---|---|---|---|---|---|---|---|---|---|---|
|  | Reward | Dev (%) | Reward | Dev (%) | Reward | Dev (%) | Reward | Dev (%) | Reward | Dev (%) | Reward | Dev (%) | Reward | Dev (%) | Reward | Dev (%) | Reward | Dev (%) | **Reward** | **Dev (%)** |
| **DQN** | 0 (0) | 86.2 (0.5) | 0 (0) | 100 (0) | 0 (0) | 100 (0) | 0 (0) | 100 (0) | 0 (0) | 100 (0) | 0 (0) | 100 (0) | 0 (0) | 100 (0) | 0 (0) | 100 (0) | 0 (0) | 100 (0) | **0.1 (0.3)** | **54 (1.4)** |
| **SA-DQN** | 30 (0) | 0 (0) | 30 (0) | 0 (0) | 20 (1.6) | 8 (10) | 30 (0) | 0 (0) | 29 (1.4) | 0.4 (0.3) | 20.8 (2.5) | 9.1 (1.1) | 30 (0) | 0 (0) | 30 (0) | 0 (0) | 20.5 (4.4) | 3 (1) | **17.3 (1.5)** | **33 (2)** |
| **DP-DQN** | 30 (0.9) | 3.5 (0.2) | 30 (0.9) | 4.5 (0.3) | 29 (1) | 3.2 (0.1) | 30.2 (1.3) | 3.7 (0.3) | 30.6 (1.4) | 4.1 (0.1) | 29.4 (1.2) | 7.3 (0.2) | 30.8 (1) | 6.5 (0.1) | 31.4 (0.8) | 7.3 (0.2) | 29 (1.1) | 10.3 (1) | **14.6 (1.5)** | **49 (1.9)** |

In response to Reviewer L64s, we have added PA-AD [2] as a new attack baseline, which is considered one of the strongest attacks in the literature. This table compares the performance of PGD, MinBest, PA-AD with budget {1/255, 3/255, 15/255}, and our attack method in Atari Freeway under DQN, SA-DQN and DP-DQN defenses. Standard deviations are reported in parentheses. The results show that our attack method achieves the best attack performance against both SA-DQN and DP-DQN in terms of both reward and deviation rate.

**Table 2-2 Reconstruction Errors and Wasserstein Distances of Different Attack Methods**
| **Freeway** | **PGD** | **1/255** | **PGD** | **3/255** | **PGD** | **15/255** | **MinBest** | **1/255** | **MinBest** | **3/255** |
|:---:|:---:|:---:|:---:|:---:|:---:|:---:|:---:|:---:|:---:|:---:|
|  | Recons. | Wass.($\times 10^{-3}$) | Recons. | Wass.($\times 10^{-3}$) | Recons. | Wass.($\times 10^{-3}$) | Recons. | Wass.($\times 10^{-3}$) | Recons. | Wass.($\times 10^{-3}$) |
| **DP-DQN** | 3.45 (0.3) | 3.1 (0.2) | 3.50 (0.3) | 7.4 (0.3) | 4.36 (0.29) | 31 (1) | 3.45 (0.3) | 3.7 (0.2) | 3.53 (0.3) | 9 (0.4) |
|  | **MinBest** | **15/255** | **PA-AD** | **1/255** | **PA-AD** | **3/255** | **PA-AD** | **15/255** | **Ours** |  |
|  | Recons. | Wass.($\times 10^{-3}$) | Recons. | Wass.($\times 10^{-3}$) | Recons. | Wass.($\times 10^{-3}$) | Recons. | Wass.($\times 10^{-3}$) | Recons. | Wass.($\times 10^{-3}$) |
| **DP-DQN** | 5.35 (0.2) | 40 (1) | 3.47 (0.29) | 4.5 (0.2) | 3.60 (0.29) | 12 (0.1) | 6.06 (0.18) | 55 (0.2) | **1.02 (0.5)** | **1.1 (0.2)** |

In response to Reviewer D5uT, we added Wasserstein distance as a new perturbation metric beyond the reconstruction error considered in the original submission. This table shows (1) the average reconstruction error (computed by our autoencoder based realism detector) of a perturbed state generated by different attacks across an episode, and (2) the average Wasserstein-1 distance between a perturbed state and the true state in the previous time step across an episode, under different attacks. The Wasserstein distance was proposed in [3] as an alternative perturbation metric to $l_p$ distances, which measures the cost of moving pixel mass and can represent image manipulations more naturally than the $l_p$ distance. We argue that reconstruction error captures static stealthiness of state perturbation, while the Wasserstein distance to the previous state captures dynamic stealthiness. The result shows that our attack method achieves both lowest reconstruction error and lowest Wasserstein distance compared with other attacks.

---

> ### Author Response · Authors · 2024-11-21
> **General Response(2)**
>
> **Table 3 Temporally Coupled Attack on Atari**
> |  | **PGD(Temporal Coupled)** |
> |:---:|:---:|
> | **Pong** | **Reward** |
> | **DQN** | -21(0) |
> | **SA-DQN** | -21(0) |
> | **DP-DQN** | 20(1.73) |
>
> In response to Reviewers eNaA and L64s, we have implemented the temporally coupled attack proposed in [4] on top of PGD (as their code is not publicly available) and evaluated the PGD-based temporally coupled attack on Atari Pong with $\epsilon = 15/255$ and $\bar{\epsilon} = 7.5/255$. This table shows that the temporally coupled attack can compromise SA-DQN but not diffusion-based DP-DQN defense even with a large perturbation budget, indicating the challenge of adapting this attack to Atari games with raw-pixel input. We conjecture that this is because the attack is still constrained by an $l_p$ norm bound, making it difficult to alter the essential semantics of image input.
>
> **Table 4-1 Pretrained Diffusion History defense against attacks in [1]**
> | Attack | Defense | Pong | FreeWay |
> |---|---|---|---|
> | **B&C** | DQN | -21 (0.00) | 23 (0.00) |
> |  | SA-DQN | 11 (0.00) | 25 (0.00) |
> |  | Diffusion History | 20 (1.41) | 27.2 (0.68) |
> | **Blur** | DQN | -21 (0.00) | 18 (0.00) |
> |  | SA-DQN | -20 (0.00) | 27 (0.00) |
> |  | Diffusion History | 20 (0.58) | 33.2 (0.37) |
> | **Rotate 1** | DQN | -20 (0.00) | 26.6 (0.45) |
> |  | SA-DQN | -18 (0.00) | 21 (0.00) |
> |  | Diffusion History | 14.6 (2.68) | 27.6 (0.45) |
> | **Shift (1,0)** | DQN | -21 (0.00) | 26 (0.00) |
> |  | SA-DQN | -21 (0.00) | 24 (0.00) |
> |  | Diffusion History | 17.8 (2.85) | 27.2 (0.37) |
>
>
> In response to Reviewer 37tk, we have implemented the four high-sensitivity direction attacks in [1] (as their code is not publicly available), and conducted new experiments to evaluate how these attacks perform under diffusion based defenses for Atari Pong and Freeway. This table shows that the Diffusion History defense with a diffusion model trained on clean data only is able to defeat the Brightness&Contrast and Blurred Observation attacks, as well as Rotation and Shifting attacks with small rotations and shifts, although it is ineffective against large rotations (>1 degree) and shifts used in [1].
>
> **Table 4-2 Fine-tuned Diffusion History against attacks in [1]**
> | Pong | Defense | Reward |
> |---|---|---|
> | **Rotate 3** | Diffusion History | 20 (0.71) |
> | **Shift (2,1)** | Diffusion History | 18.8 (1.79) |
>
> We further fine-tuned a diffusion model by randomly applying rotations and shifts to the game frames during training, where the rotation degree is randomly chosen between 0 and 3, and the shift magnitude is randomly chosen between (0,0) and (3,3). This table shows that the fine-tuned Diffusion History defense can successfully mitigate both Rotation and Shifting attacks, even under relatively large rotations and shifts considered in [1]. In contrast, the Diffusion History defense is ineffective against our policy-adaptive attack.
>
> **Table 4-3 Wasserstein Distances of attacks in [1] and ours**
> | **Freeway**  | **Wasserstein** |
> |:---:|:---:|
> | **B&C** | 0.036(0.004) |
> | **Blur** | 0.006(0.003) |
> | **Rotate 1** | 0.006(0.004) |
> | **Shift (1,0)** | 0.07(0.001) |
> | **Ours** | 0.001(0.0002) |
>
> This table compares the average Wasserstein Distance between a perturbed state and the previous step’s true state across an episode, under the attack methods in [1] and our attack. The results show that our attack method has the lowest Wasserstein distance compared with the four attacks evaluated in [1], indicating that our attack is more stealthy.
>
> **Table 5 DDPM vs. EDM**
> | **Pong** | **DDPM** |  |  | **EDM** |  |  |
> |:---:|:---:|:---:|:---:|:---:|:---:|:---:|
> |  | Reward | Manipulation Rate(%) | Deviation Rate(%) | Reward | Manipulation Rate(%) | Deviation Rate(%) |
> | **DQN** | -20.6(0.5) | 76.6(1) | 83.6(1) | -20.7(0.5) | 87.1(1.9) | 89.6(1.7) |
> | **Diffusion History** | 5.4(5.6) | 15.1(0.4) | 45.2(0.3) | 6.0(6.2) | 8.4(0.5) | 25.3(0.9) |
> | **Running Time** | ~5 sec |  |  | ~0.2 sec |  |  |
>
> As recommended by Reviewer L64s, we have compared DDPM and EDM in terms of effectiveness and efficiency. The results in the table show that EDM and DDPM exhibit similar attack performance. However, DDPM is significantly slower than EDM in terms of running time (the average time needed to generate a single perturbed state during testing), making DDPM incapable of generating real-time attacks during testing. This validates the selection of EDM as the diffusion model for constructing our attacks.
>
> [1] Ezgi Korkmaz. Adversarial Robust Deep Reinforcement Learning Requires Redefining Robustness. AAAI 2023.
>
> [2] Sun, Y., Zheng, R., Liang, Y., & Huang, F. Who Is the Strongest Enemy? Towards Optimal and Efficient Evasion Attacks in Deep RL. ICLR 2022.
>
> [3] Eric Wong, Frank R. Schmidt, and J. Zico Kolter. Wasserstein Adversarial Examples via Projected Sinkhorn Iterations. ICML 2019.
>
> [4] Liang et al., Game-Theoretic Robust Reinforcement Learning Handles Temporally-Coupled Perturbations. ICLR 2024

---

### Author Response · Authors · 2024-11-25
**Looking forward to your feedback**

Dear Reviewers:

Thank you once again for your insightful and helpful reviews. As the rebuttal phase is coming to a close, we are eager to know if our responses have satisfactorily addressed your concerns. If there is any additional information or clarification we can provide, please do not hesitate to let us know. Thank you so much for your time!

---

### Meta-Review · Area_Chair_xj5j · 2024-12-22

**Metareview:**

The paper discusses a novel adversarial attack to RL agents, by creating realistic perturbations using diffusion models. On the positive side, this attack is novel and can generate perturbed states that are semantically different from the true states while remaining realistic to avoid detection. However, it is difficult to quantitatively judge the realism and stealthiness of the proposed attack, as these terms do not have a precise mathematical definition. The evaluation results are not surprising since it is expected that many existing defenses built on Lp norm perturbation are not robust against the proposed attack (which can have a large norm and actually change the semantics). The experiments are not comprehensive enough (lacking environments beyond 3 easy Atari games and ablation studies). Considering these factors, the current form of this paper cannot be accepted at ICLR.

**Additional Comments On Reviewer Discussion:**

The authors provided detailed responses during the discussion period, and the AC has checked them carefully. The initial version of the paper lacks discussion of a significant amount of related work. The paper was updated to include missing references that should have been discussed in the paper. However, the key weaknesses of the paper remain. Especially multiple reviewers (eNaA, D5uT) questioned this attack setting, and the AC shared the same concern. Also, although several new tables were provided as new results during the discussion period, the results are not comprehensive enough compared to most other published work in this field.

---

### Decision · Program_Chairs · 2025-01-22

Reject